# Codon usage biases co-evolve with transcription termination machinery to suppress premature cleavage and polyadenylation

**Zhipeng Zhou[1†], Yunkun Dang[2,3†]\*, Mian Zhou[4], Haiyan Yuan[1], Yi Liu[1]\***

[1]Department of Physiology, The University of Texas Southwestern Medical Center, Dallas, United States; [2]State Key Laboratory for Conservation and Utilization of Bio-Resources in Yunnan, Yunnan University, Kunming, China; [3]Center for Life Science, School of Life Sciences, Yunnan University, Kunming, China; [4]State Key Laboratory of Bioreactor Engineering, East China University of Science and Technology, Shanghai, China

**\*For correspondence:**
ykdang@ynu.edu.cn (YD);
yi.liu@utsouthwestern.edu (YL)

[†]These authors contributed equally to this work

**Competing interests:** The authors declare that no competing interests exist.

**Abstract** Codon usage biases are found in all genomes and influence protein expression levels. The codon usage effect on protein expression was thought to be mainly due to its impact on translation. Here, we show that transcription termination is an important driving force for codon usage bias in eukaryotes. Using *Neurospora crassa* as a model organism, we demonstrated that introduction of rare codons results in premature transcription termination (PTT) within open reading frames and abolishment of full-length mRNA. PTT is a wide-spread phenomenon in *Neurospora,* and there is a strong negative correlation between codon usage bias and PTT events. Rare codons lead to the formation of putative poly(A) signals and PTT. A similar role for codon usage bias was also observed in mouse cells. Together, these results suggest that codon usage biases co-evolve with the transcription termination machinery to suppress premature termination of transcription and thus allow for optimal gene expression.
DOI: https://doi.org/10.7554/eLife.33569.001

## Introduction

Due to the redundancy of triplet genetic codons, most amino acids are encoded by two to six synonymous codons. Synonymous codons are not used with equal frequencies, a phenomenon called codon usage bias (*Ikemura, 1985*; *Sharp et al., 1986*; *Comeron, 2004*; *Plotkin and Kudla, 2011*; *Hershberg and Petrov, 2008*). Highly expressed proteins are mostly encoded by genes with preferred codons, and codon optimization has been routinely used to enhance heterologous protein expression. In addition, positive correlations between codon usage and protein expression levels are observed in different organisms (*Hiraoka et al., 2009*; *Duret and Mouchiroud, 1999*). These results suggest that codon usage plays an important role in regulating gene expression levels. Efficient and accurate translation was thought to be a major selection force for codon usage biases (*Hiraoka et al., 2009*; *Duret and Mouchiroud, 1999*; *Akashi, 1994*; *Drummond and Wilke, 2008*; *Xu et al., 2013*; *Zhou et al., 2013a*; *Lampson et al., 2013*; *Pershing et al., 2015*). Recent studies also demonstrated that codon usage affects co-translational protein folding by regulating translation elongation rate in both prokaryotes and eukaryotes (*Zhou et al., 2013a*; *Spencer et al., 2012*; *Pechmann et al., 2014*; *Yu et al., 2015*; *Zhou et al., 2015*; *Fu et al., 2016*; *Zhao et al., 2017*).

Although the correlation between codon usage and gene expression level can be explained by translation efficiency, recent studies suggest that overall translation efficiency is mainly determined

by translation initiation, a process that is mostly determined by RNA structure but not codon usage near the translational start site (*Kudla et al., 2009*; *Pop et al., 2014*; *Tuller et al., 2010*). In addition, codon usage was found to be an important determinant of RNA levels in many organisms (*Presnyak et al., 2015*; *Boël et al., 2016*; *Zhou et al., 2016*; *Kudla et al., 2006*; *Krinner et al., 2014*). In some organisms, codon usage was shown to affect RNA stability (*Presnyak et al., 2015*; *Boël et al., 2016*; *Zhou et al., 2016*; *Kudla et al., 2006*; *Krinner et al., 2014*; *Mishima and Tomari, 2016*; *Bazzini et al., 2016*). In *Neurospora* and mammalian cells, codon usage has also been shown to be an important determinant of gene transcription levels (*Zhou et al., 2016*; *Newman et al., 2016*). Therefore, codon usage can regulate gene expression beyond the translation process.

Transcription termination is a critical process in regulating gene expression. In eukaryotes, the maturation of mRNA is a two-step process involving endonucleolytic cleavage of the nascent RNA followed by the synthesis of the polyadenosine (poly(A)) tail (*Tian and Graber, 2012*; *Shi and Manley, 2015*; *Proudfoot, 2011*; *Proudfoot, 2016*; *Tian and Manley, 2017*; *Kuehner et al., 2011*; *Porrua and Libri, 2015*). The polyadenylation sites, also known as the poly(A) sites or pA sites, are defined by surrounding *cis*-elements recognized by multiple proteins. These *cis*-elements, called poly(A) signals, are generally AU-rich sequences and have conserved nucleotide composition in eukaryotes. In mammals, the hexamer AAUAAA (or other close variants), referred as polyadenylation signal (PAS), is one of the most prominent poly(A) signals. Other *cis*-elements, such as upstream U-rich elements, downstream U-rich element, and downstream GU-rich element, also play important roles in the transcription termination process. In yeast, poly(A) signals include an upstream efficiency element (EE), an upstream position element (PE), which is equivalent to PAS in mammals, and two U-rich elements flanking the poly(A) sites (*Moqtaderi et al., 2013*; *Mata, 2013*; *Ozsolak et al., 2010*; *Schlackow et al., 2013*; *Liu et al., 2017a*). Mutation of these poly(A) signals impairs the efficiency of transcription termination and leads to defect in mRNA processing (*Tian and Graber, 2012*; *Shi and Manley, 2015*; *Proudfoot, 2011*; *Proudfoot, 2016*; *Tian and Manley, 2017*; *Kuehner et al., 2011*; *Porrua and Libri, 2015*).

Although most of the transcriptional events terminate in 3′ untranslated region (3′ UTR) of protein-coding genes, premature transcription termination (PTT) also occurs in 5′ UTR, intron and exon, which is also referred as premature cleavage and polyadenylation (PCPA) (*Tian et al., 2007*; *Kaida et al., 2010*; *Berg et al., 2012*; *Jan et al., 2011*; *Liu et al., 2017b*; *Ulitsky et al., 2012*; *Yang et al., 2016*). For example, premature transcription termination occurs in the intron of the *Arabidopsis* FCA gene and is involved in the control of flowering timing (*Quesada et al., 2003*; *Macknight et al., 2002*). Moreover, PCPA in intron is a conserved regulatory mechanism for CstF-77 gene from fly to human (*Mitchelson et al., 1993*; *Pan et al., 2006*; *Luo et al., 2013*). Recent genome-wide studies have shown that PCPA within introns is a widespread phenomenon in eukaryotes (*Tian et al., 2007*; *Liu et al., 2017b*; *van Hoof et al., 2002*; *Frischmeyer et al., 2002*; *Mayr and Bartel, 2009*). In addition, PCPA also occurs within coding regions (*Tian et al., 2007*; *Kaida et al., 2010*; *Jan et al., 2011*; *Dunlap and Loros, 2017*). It has been shown that the expression of heterologous gene can be suppressed due to PCPA in coding regions (*Diehn et al., 1998*; *Tokuoka et al., 2008*). PCPA also occurs in coding regions of endogenous genes in both yeast and human (*van Hoof et al., 2002*; *Frischmeyer et al., 2002*; *Georis et al., 2015*). Importantly, many poly(A) sites have been mapped to coding regions using poly(A) sequencing methods (*Jan et al., 2011*; *Liu et al., 2017b*; *Ulitsky et al., 2012*; *Yang et al., 2016*). Codon optimization has been previously shown to increase heterologous gene expression in *Aspergillus oryzae* (*Tokuoka et al., 2008*). However, the effect of codon usage on premature transcription termination of endogenous genes is not clear.

The filamentous fungus *Neurospora crassa* exhibits a strong codon usage bias for C or G at wobble positions and has been an important model organism studying the roles of codon usage biases (*Zhou et al., 2013a*; *Yu et al., 2015*; *Zhou et al., 2015*; *Radford and Parish, 1997*).In *Neurospora*, codon usage is a major determinant of gene expression levels and correlates strongly with protein and RNA levels (*Zhou et al., 2016*). We showed previously that codon usage can regulate mRNA levels at the level of transcription by influencing chromatin structure (*Zhou et al., 2016*). In this study, we showed that premature transcription termination within open reading frames is affected by codon usage bias. Moreover, a similar phenomenon is observed in mouse, another C/G-biased organism. Therefore, in addition to effects on translation, transcription termination serves as a conserved driving force in shaping codon usage biases in C/G-biased organisms.

## Results

### Codon deoptimization of the amino-terminal end of the *frq* open reading frame abolishes the production of full-length mRNA

We previously showed that codon optimization of circadian clock gene *frequency* (*frq*) leads to changes in FRQ expression level and protein structure (*Zhou et al., 2013a*; *Zhou et al., 2015*). To determine the impact of non-optimal codons on FRQ expression, we codon deoptimized the amino-terminal end of *frq* ORF (amino acids 12–163) by replacing the wild-type codons with non-optimal synonymous codons (*Figure 1A*). In the *frq*-deopt1 construct, 59 codons were replaced by non-optimal codons. In the *frq*-deopt2 construct, 98 codons were replaced by the least preferred codons (*Figure 1—figure supplement 1*). These two constructs and the wild-type *frq* (wt-*frq*) construct were transformed individually into an *frq* knock-out strain (*frq*KO) at the *his-3* locus by homologous recombination (*Aronson et al., 1994a*). In the strains expressing the wild-type *frq* construct, the production of conidia (asexual spore) was rhythmic with a period of about 22 hr (*Figure 1B*). However, the conidiation rhythm of the strains expressing the two codon-deoptimized *frq* constructs was abolished, indicating that the deoptimized *frq* genes are not functional (*Figure 1B*). Surprisingly, no FRQ expression was detected in either of the deoptimized strains by western blot (*Figure 1C*). Northern blot and strand-specific RT-qPCR using a set of primers targeting the middle region of *frq* ORF showed that no full-length *frq* mRNA was produced in the deoptimized strains (*Figure 1D and E*). Together, these results indicate that the introduction of rare synonymous codons within this region of *frq* abolishes the expression of full-length *frq* mRNA.

### Codon deoptimization of *frq* results in premature cleavage and polyadenylation

We have previously shown that rare codons can result in gene silencing through histone H3 trimethylation at lysine 9 (H3K9me3), and the wild-type *frq* locus is marked by H3K9me3 (*Zhou et al., 2016*; *Dang et al., 2013*; *Belden et al., 2011*). Thus, we examined whether the loss of *frq* expression in the codon deoptimized strains was due to an increase of H3K9me3 at the *frq* locus. Chromatin immunoprecipitation (ChIP) assay using an H3K9me3 antibody, however, showed that the H3K9me3 levels at the *frq* locus were comparable in the wild-type *frq* and *frq*-deopt2 strains (*Figure 2—figure supplement 1A and B*), suggesting that the loss of full-length *frq* mRNA in the deoptimized *frq* is not due to H3K9me3-mediated transcriptional silencing. Transcription of *frq* is activated by the binding of the complex of White Collar-1 (WC-1) and White Collar-2 (WC-2) to the *frq* promoter, and the expression of FRQ inhibits WC binding (*Heintzen and Liu, 2007*; *Dunlap, 2006*). A ChIP assay showed that WC-2 binding at the *frq* promoter was elevated in the *frq*-deopt2 strain (*Figure 2—figure supplement 1C*), suggesting that the loss of full-length *frq* mRNA expression is not due to transcriptional gene silencing. Consistent with this result, strand-specific RT-qPCR using a set of primers targeted to an intronic region in the 5' UTR of *frq* showed that the *frq* pre-mRNA was increased significantly in the *frq*-deopt2 strain (*Figure 2—figure supplement 1D*). These results indicate that even though full-length *frq* mRNA could not be detected in the codon deoptimized strains, the transcription of *frq* was actually significantly increased.

Since *frq* RNA can be detected by strand-specific RT-qPCR using primers targeted to 5' UTR but not to a region of the *frq* ORF that is downstream of the codon deoptimized region, we hypothesized that codon deoptimization resulted only in truncated *frq* mRNA. To test this hypothesis, we performed northern blot analysis using a probe targeted to the 5' end of *frq* mRNA (*Figure 2—figure supplement 1E*). As expected, truncated *frq* mRNAs but not full-length *frq* mRNA were detected in both of the deoptimized *frq* strains (*Figure 2A*). To characterize the nature of these truncated *frq* mRNAs, we first examined whether these RNAs were polyadenylated by purifying poly(A) RNAs using oligo-dT beads. Like the full-length *frq* mRNA, the truncated *frq* mRNA species were also enriched after oligo-dT purification (*Figure 2B*). To further confirm these results, we performed poly(A) tail-based 3' RACE (*Scotto-Lavino et al., 2006*) and mapped the 3' ends of the truncated mRNA species in the *frq*-deopt1 and *frq*-deopt2 strains. We observed a cluster of 3' ends within the deoptimized region (112-141nt downstream of the start codon) of *frq* genes (*Figure 2C*). Interestingly, PAS variants (AUAAAU in the *frq*-deopt1 and AUAAAA in the *frq*-deopt2), which were located

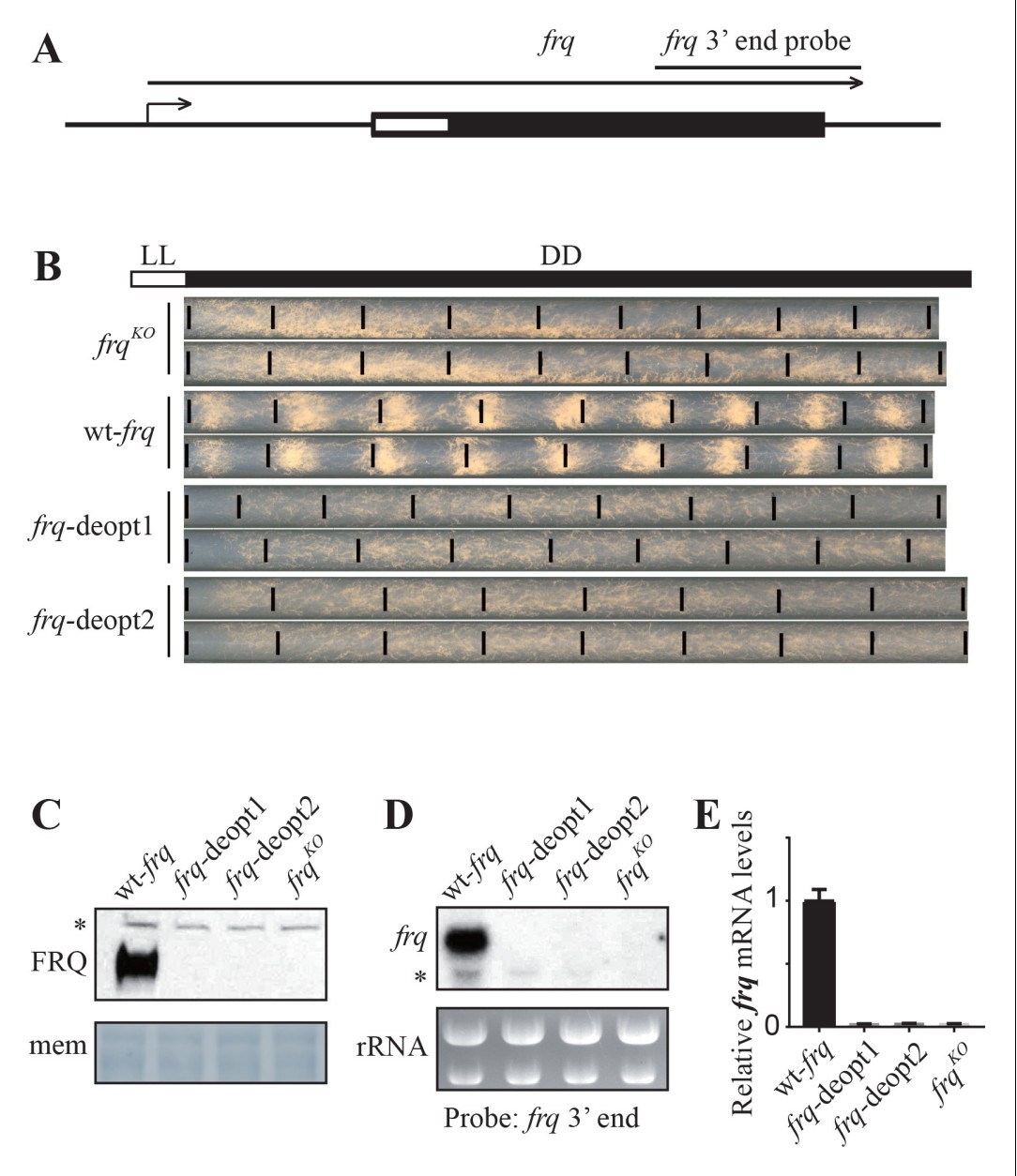

**Figure 1.** Codon deoptimization of the amino-terminal end of the *frq* ORF abolished the expression of full-length *frq* mRNA. (**A**) A diagram showing the *frq* locus. (**B**) Race tube analysis showing the conidiation rhythm of the *frq*[KO], wt-*frq*, *frq*-deopt1, and *frq*-deopt2 strains. The strains were first cultured in constant light (LL) for 1 day before transferred to constant darkness (DD). Black lines mark the growth fronts every 24 hr. The distance between asexual spore bands was measured and then divided by growth rate to calculate the period length of conidiation rhythm. For the wt-*frq* strain, the period of conidiation rhythm was 22.07 ± 0.04 hr. (**C**) Western blot showing FRQ protein levels in *frq*[KO], wt-*frq*, *frq*-deopt1, and *frq*-deopt2 strains. (**D**) Northern blot showing the expression of full-length *frq* mRNA in the indicated strains. An RNA probe specific for 3' end of *frq* was used in this experiment. (**E**) Strand-specific RT-qPCR results showing *frq* mRNA levels in the indicated strains. Primers used for the qPCR were targeted to the middle of *frq* ORF.

DOI: https://doi.org/10.7554/eLife.33569.002

The following figure supplement is available for figure 1:

**Figure supplement 1.** DNA sequences of 5' end of the *frq* open reading frame region for the indicated constructs.

DOI: https://doi.org/10.7554/eLife.33569.003

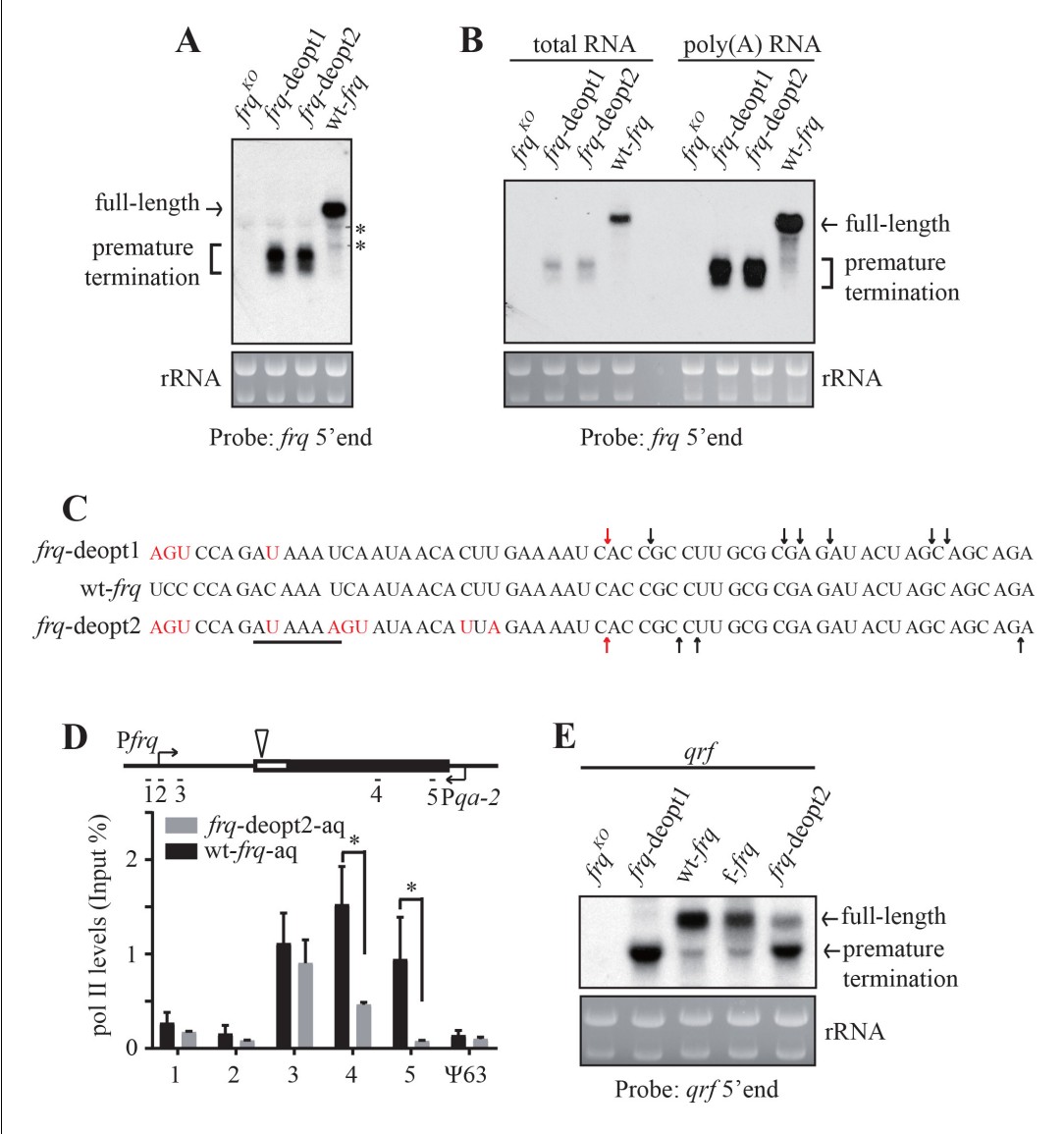

**Figure 2.** Codon deoptimization of *frq* results in premature transcription termination. (**A**) Northern blot showing the presence of truncated *frq* mRNA species in both de-optimized strains using an RNA probe targeted to 5' end of *frq* mRNA (indicated in ; *Figure 2—figure supplement 1E*). * indicates a non-specific band. (**B**) Northern blot showing both full-length and truncated *frq* mRNA are enriched in poly(A)-containing RNAs. Poly(A) RNAs were purified from total RNAs by using oligo-dT beads. Equal amounts of total RNA or poly(A) RNA were loaded in each lane. Probe specific for 5' end of *frq* was used. (**C**) Poly(A) sites mapped by 3' RACE. Arrows indicate the mapped poly(A) sites, the red arrows indicate the major poly(A) site that was found in both *frq*-deopt1 and *frq*-deopt2 strains, and the black line indicates potential PAS motif (AUAAAU in *frq*-deopt1 and AAUAAA in *frq*-deopt2). Nucleotides that are mutated are shown in red. (**D**) ChIP assay showing RNA pol II levels at the *frq* transgene loci in the wt-*frq*-aq and *frq*-deopt2-aq strains. The ChIP results were normalized by input DNA and represented as Input%. The promoter of *qrf* was replaced by a *qa-2* promoter and tissue were cultured in the absence quinic acid to block *qrf* transcription. The triangle on the top indicates the location of mapped poly(A) sites. The previously known heterochromatin region ψ63 in *Neurospora* was used as the negative control. Error bars shown are standard deviations (n = 3). *p<0.05. (**E**) Northern blot analysis showing premature transcription termination of *qrf*. f-*frq* is an *frq* codon-optimized strain (*Zhou et al., 2013a*).
DOI: https://doi.org/10.7554/eLife.33569.004
The following figure supplement is available for figure 2:

**Figure supplement 1.** Mechanism of the codon usage-mediated gene expression changes at the frq locus
DOI: https://doi.org/10.7554/eLife.33569.005

in 18-nt upstream of mapped poly(A) sites, were created by codon deoptimization of *frq*, suggesting these truncated mRNAs may be produced by PAS-dependent pathway (*Figure 2C*).

There are two possibilities for how these truncated polyadenylated *frq* mRNAs can be produced: PAS-dependent premature transcription termination or partial degradation of full-length *frq* mRNAs followed by polyadenylation (*van Hoof et al., 2002*; *Frischmeyer et al., 2002*; *West et al., 2006*; *LaCava et al., 2005*). In the case of premature transcription termination, RNA polymerase II (pol II) terminates after synthesis of the 5' region of the pre-mRNA, which is then released from the chromatin (*Proudfoot, 2016*). It should be noted that *frq* locus is not only transcribed from sense direction to produce *frq* mRNA, it is also transcribed from antisense direction to generate the long non-coding RNA *qrf* (*Kramer et al., 2003*; *Xue et al., 2014*) (*Figure 2—figure supplement 1E*), which can complicate the interpretation of the ChIP results. To overcome this complication, we created two additional *frq* constructs, wt-*frq*-aq, and *frq*-deopt2-aq, in which the promoter of *qrf* was replaced by the quinic acid (QA) inducible *qa-2* promoter. In *frq* null strains transformed with these constructs, expression of the full-length and truncated *frq* was not dependent on QA, but *qrf* was only expressed in the presence of QA (*Figure 2—figure supplement 1F*). Therefore, we cultured wt-*frq*-aq and *frq*-deopt2-aq strains in the absence of QA and performed a ChIP assay using an antibody against pol II phosphorylated at serine 2. The pol II levels at the *frq* promoter and 5' UTR were comparable in the wt-*frq*-aq and *frq*-deopt2-aq strains, but pol II levels in the middle and 3' end of *frq* ORF were decreased dramatically in the *frq*-deopt2-aq strain compared to the wt-*frq*-aq strain (*Figure 2D*). Together, these results demonstrate that codon deoptimization of *frq* abolished its expression due to premature transcription termination.

Codon deoptimization of *frq* also resulted in the premature transcription termination of *qrf* as indicated by the loss of full-length *qrf* and appearance of truncated *qrf* mRNA in the *frq*-deopt1 and *frq*-deopt2 strains (*Figure 2E* and *Figure 2—figure supplement 1F*). 3' RACE result showed that the 3' ends of the truncated *qrf* mRNAs in the *frq*-deopt1 strains also localized in the deoptimized region with a potential PAS (AUAAAA) motif 21-nt upstream of the 3' ends (*Figure 2—figure supplement 1G*). It should be noted that the wt-*frq* gene also has the same putative PAS motif, suggesting that the nucleotide sequence near PAS motif is also required for transcription termination.

## PAS motif and other *cis*-elements created by codon deoptimization are important for premature transcription termination of *frq*

The results above suggest that codon deoptimization of *frq* may create potential poly(A) signals that can result in premature transcription termination of *frq*. To identify the codon or codons that are critical for premature transcription termination, we create additional codon deoptimized *frq* genes (*frq*-deopt3, 4, and 5) by deoptimizing different regions of *frq* ORF around the 3' ends identified in the *frq*-deopt2 strains (*Figure 3A*). Neither full-length *frq* mRNA nor FRQ protein was detected in the *frq*-deopt3 strain (*Figure 3B and C*), suggesting that the deoptimized region in *frq*-deopt3 contains all elements sufficient to trigger transcription termination. The low level of the prematurely terminated products in the *frq*-deopt3 strain, suggesting that these products may be rapidly degraded by the RNA quality control mechanisms (*van Hoof et al., 2002*; *Frischmeyer et al., 2002*; *Doma and Parker, 2007*; *Vanacova and Stefl, 2007*; *Schmid and Jensen, 2010*). In the *frq*-deopt4 strain, both full-length *frq* RNA and FRQ protein were detected, but their levels were significantly lower than that in the wt-*frq* strain (*Figure 3B and C*). ChIP result showed that polII levels at the *frq* transcription start site were comparable in the wt-*frq* and *frq*-deopt4 strains (*Figure 3—figure supplement 1B*), suggesting that the decrease of full-length *frq* mRNA in the *frq*-deopt4 strain was not due to transcriptional silencing. Notably, the level of premature terminated *frq* RNAs in the *frq*-deopt4 strain was also lower than that in the *frq*-deopt2 strain, suggesting that transcription termination efficiency was decreased due to the lack of some elements. The levels of *frq* mRNA and FRQ protein in the *frq*-deopt5 strain were higher than those in the *frq*-deopt4 strain but were much lower than those in the wt-*frq* strain (*Figure 3B and C*). Premature termination products in the *frq*-deopt5 strain were further decreased compared to that in the *frq*-deopt4 strain. Although *frq*-deopt4 and *frq*-deopt5 strain share the same PAS motif, the production of premature termination products in these strains was markedly reduced, suggesting that other *cis*-elements surrounding the PAS motif are also important for PCPA efficiency.

Sequence analysis revealed that a GAC to GAU mutation created a potential PAS motif in both *frq*-deopt1 and *frq*-deopt2 constructs (*Figure 2C*). Another UCC to AGU mutation 4-nt upstream of

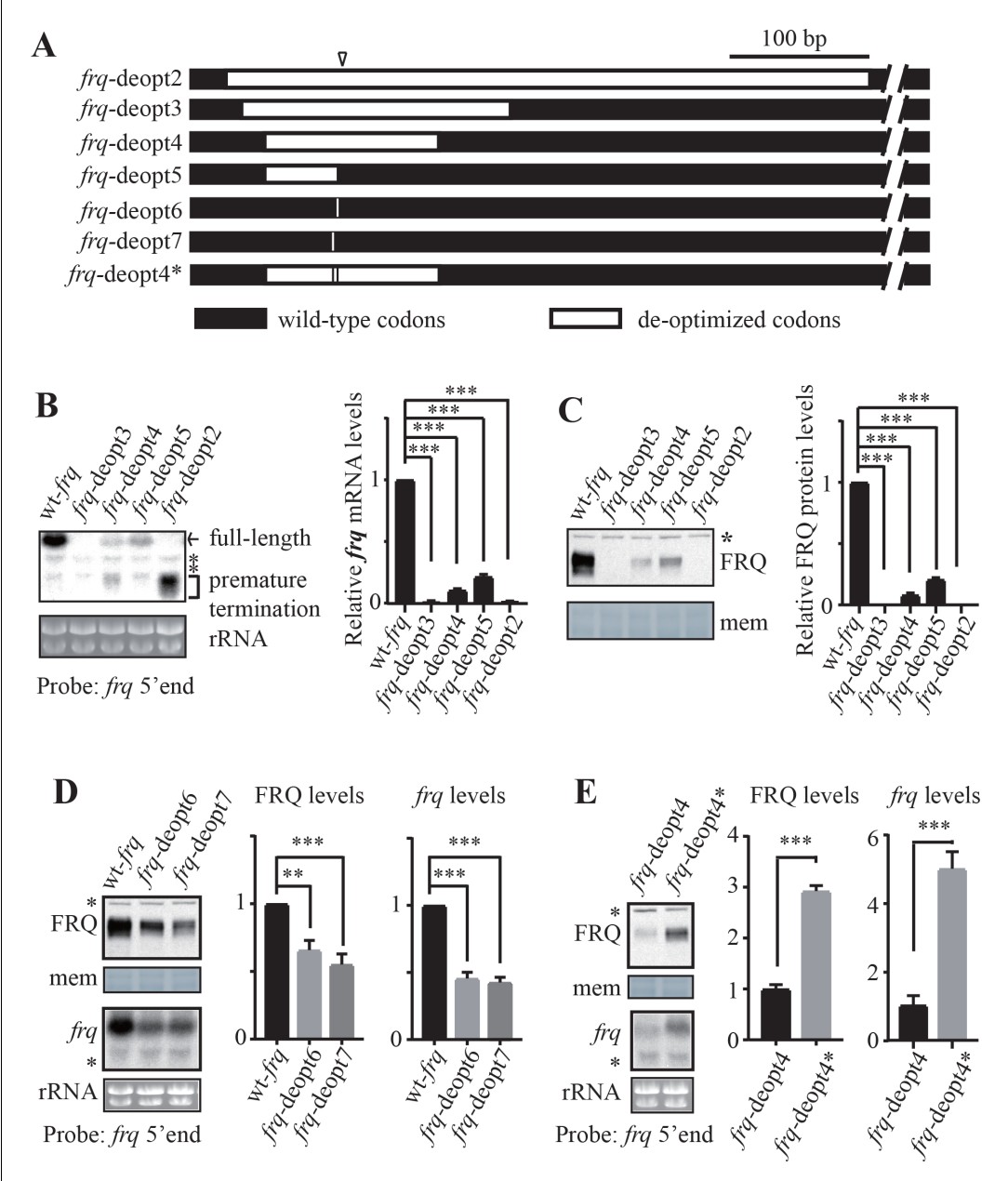

**Figure 3.** Rare codons promote while optimal codons suppress premature transcription termination of *frq*. (A) A diagram showing the constructs created to map codons important for premature transcription termination. The triangle indicates the location of the mapped poly(A) sites. Black bars indicate the regions where wild-type codons are used, whereas white bars indicate regions that are codon de-optimized. (B) Left panel, northern blot analysis showing the expression of full-length and premature terminated *frq* mRNA species in the indicated strains. The asterisks indicate non-specific bands. A probe for *frq* 5' end was used. Right panel, densitometric analyses of results from three independent experiments. Error bars shown are standard deviations (n = 3). ***p<0.001. (C) Left panel, western blot result showing FRQ protein levels in the indicated strains. The asterisk indicates a non-specific band. Right panel, densitometric analyses of results from three independent experiments. (D) Left top panel, western analyses showing FRQ protein levels in the wt-*frq*, *frq*-deopt6, and *frq*-deopt7 strains. Left bottom panel, northern blot showing full-length *frq* mRNA levels in the indicated strains. Middle panel, densitometric analysis of FRQ levels from three independent experiments. Right, densitometric analyses of full-length *frq* mRNA levels from three independent experiments. Error bars shown are standard deviations (n = 3). **p<0.01, ***p<0.001. (E) Left top panel, western analyses showing FRQ protein levels in the *frq*-deopt4 and *frq*-deopt4* strains. Left bottom panel, northern blot showing full-length *frq* mRNA levels in the indicated strains. An RNA probe specific for 5' end of *frq* was used. Middle, densitometric analyses of FRQ levels from three independent experiments. Right, densitometric analyses of full-length *frq* mRNA levels from three independent experiments. Error bars shown are standard deviations (n = 3). ***p<0.001.

DOI: https://doi.org/10.7554/eLife.33569.006

*Figure 3 continued on next page*

*Figure 3 continued*

The following figure supplement is available for figure 3:

**Figure supplement 1.** Codon deoptimization of *frq* abolish circadian clock function and reduction of *frq* expression.

DOI: https://doi.org/10.7554/eLife.33569.007

this PAS motif was also present in both *frq*-deopt1 and *frq*-deopt2 constructs. Thus, we hypothesized that deoptimization of these codons triggers premature termination of *frq* transcription. To test this hypothesis, we made two more constructs in which only one codon in the wild-type *frq* gene was deoptimized: *frq*-deopt6, in which the GAC codon in the wt-*frq* was replaced by GAU and therefore created PAS motif AUAAAA, and *frq*-deopt7, in which the UCC codon was changed to AGU (*Figure 3A*). Even though the effect of these mutations on *frq* mRNA levels was not as dramatic as for the other constructs, these constructs with only a single synonymous codon mutation resulted in significant reduction of *frq* mRNA and FRQ protein levels and the loss of circadian conidiation rhythms (*Figure 3D* and *Figure 3—figure supplement 1A and C*). To further confirm the importance of PAS motif, we created the *frq*-deopt4* construct, in which the PAS motif in the *frq*-deopt4 was mutated by replacing these two non-preferred codons with the synonymous optimal codons. We found that both *frq* RNA and FRQ levels increased significantly in the *frq*-deopt4* strain compared to that in the *frq*-deopt4 strain, and the level of prematurely terminated *frq* mRNA was also reduced (*Figure 3E* and *Figure 3—figure supplement 1D*). These results indicate that the PAS motif created by codon deoptimization is important for PCPA in *frq* and additional motifs surrounding the PAS are also involved in promoting PCPA. Together these results suggest that codon usage, by affecting the formation of potential PAS and other *cis*-elements, plays an important role in suppressing premature transcription termination to allow production of full-length mRNAs in *Neurospora*.

## Genome-wide identification of premature transcription termination events in the open reading frames of *Neurospora* genes

Our results suggest that the use of rare codons in *Neurospora* genes may play a role in promoting premature transcription termination. Thus, we asked whether PCPA also occur in endogenous genes by performing genome-wide identification of poly(A) sites. Because premature transcription termination products are usually rapidly degraded in the cytosol (*van Hoof et al., 2002*; *Frischmeyer et al., 2002*), we extracted polyadenylated RNAs from nuclei and mapped the 3′ ends using a modified poly(A)-tail-primed sequencing method (2P-seq) (*Spies et al., 2013*). To limit false positive reads, we used a low concentration of reverse transcription (RT) primer during sequencing library preparation (*Scotto-Lavino et al., 2006*) and a stringent filtering procedure for data analysis (see Materials and methods). It should be noted that the poly(A) sites identified by this method may not be only limited from those generated by the PAS-dependent pathway (*Porrua and Libri, 2015*; *West et al., 2006*; *LaCava et al., 2005*). Two biological replicates of 2P-seq results were highly consistent (*Figure 4—figure supplement 1A*), indicating the reliability of the method. Approximately 20 million reads were generated from two independent samples. The vast majority of these reads were mapped to 3′ UTRs of genes (i.e. annotated 3′ UTR plus 1000 nt downstream) and the rest of reads were mapped to ORFs, introns, 5′ UTRs, and intergenic regions. We focused our analysis on the reads mapped to the 3′ UTRs and ORFs, with deduced poly(A) sites referred as 3′ UTR-pA and ORF-pA, respectively. The putative PAS regions (including −30 ~ −10nt A-rich region and −10 ~+10 nt U-rich region) are referred as 3′ UTR-PAS and ORF-PAS, respectively. From 9795 annotated genes, 7755 genes have 3′ UTR-PAS signals (RPM >1), which is comparable to our previous RNA-seq results (*Zhou et al., 2016*), indicating the sensitivity of our 2P-seq method. Within these genes, 4557 genes were identified to have at least 1 ORF-pA signal. The mapped 2P-seq reads for two *Neurospora* genes (*NCU09435* and *NCU00931*), which had considerable reads in their ORFs, are shown in *Figure 4A* and *Figure 4—figure supplement 1B*. To confirm these results, northern blot analyses were performed. As expected, in addition to the full-length mRNAs, small amounts of prematurely terminated mRNA species with expected sizes based on the location of the 2P reads were also detected (*Figure 4B*). These results suggest that premature transcription termination within ORFs is a common phenomenon in *Neurospora*.

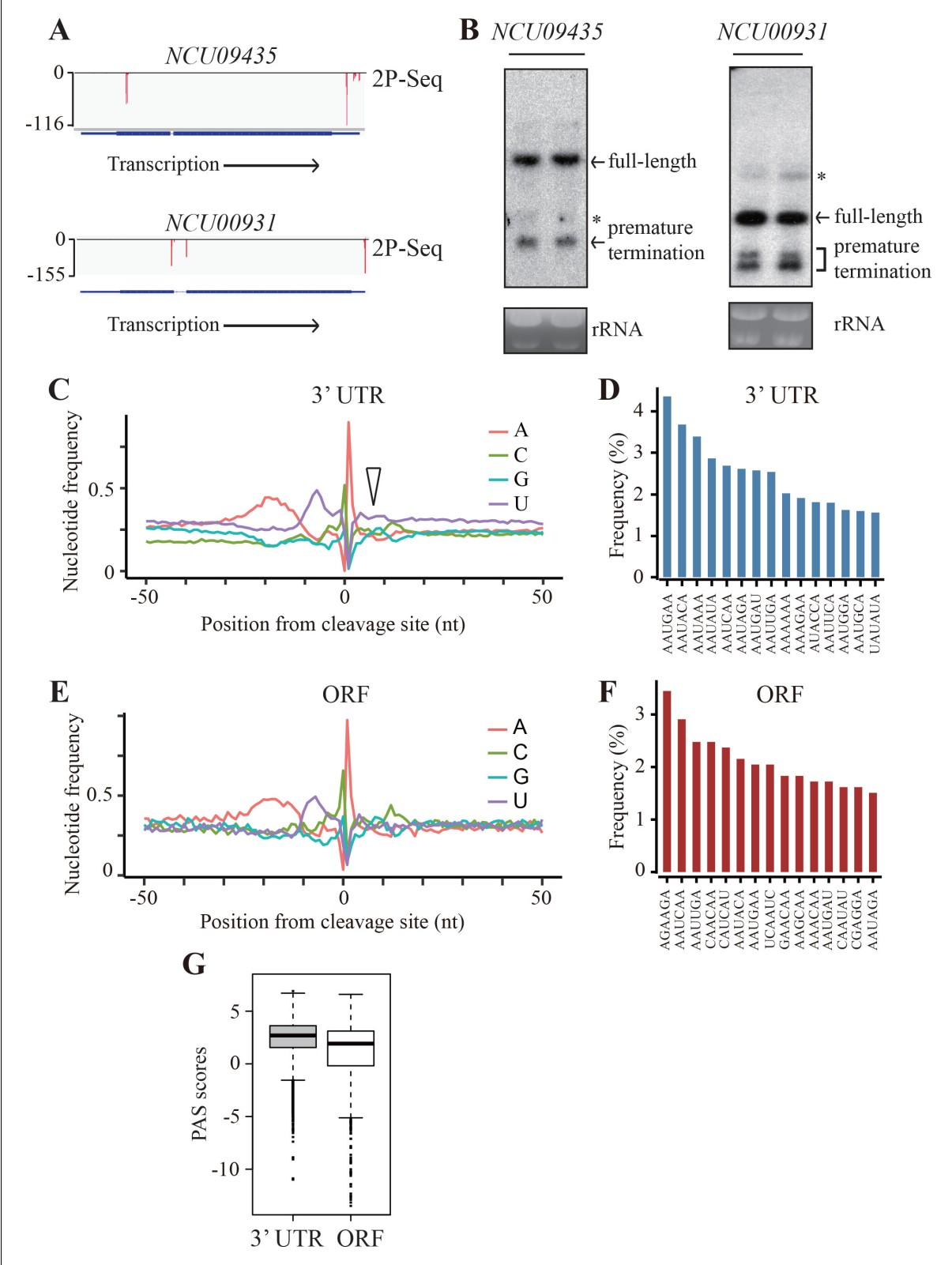

**Figure 4.** Genome-wide identification of premature transcription termination events in ORF of endogenous *Neurospora* genes. (**A**) 2P-seq results on *NCU09435* (top) and *NCU00931* (bottom) genes showing the transcription termination events in the 3' UTR and ORF. (**B**) Northern blot analyses showing the presence of both full-length and prematurely terminated *NCU09435* mRNA (left) and *NCU00931* mRNA (right) in the wild-type strain, respectively. An RNA probe specific for 5' end of *NCU09435* or *NCU00931* was used, respectively. * indicates a non-specific band. (**C**) Genome-wide

*Figure 4 continued on next page*

*Figure 4 continued*

nucleotide composition surrounding mRNA 3' ends in the 3'UTRs. 0 indicates the position of the mapped 3' end of mRNA. The triangle indicates the downstream U-rich element. (**D**) Top 15 most frequently used PAS motifs found in the A-rich element of 3' UTR-PAS. (**E**) Genome-wide nucleotide sequence composition surrounding ORF-pA sites. (**F**) Top 15 most frequently used PAS motifs found in the A-rich element of ORF-PAS. (**G**) Box-plots of PAS scores for 3'UTR-PAS and ORF-PAS determined in *Neurospora*.

DOI: https://doi.org/10.7554/eLife.33569.008

The following figure supplement is available for figure 4:

**Figure supplement 1.** Analyses of the 2P-seq results in *Neurospora*.

DOI: https://doi.org/10.7554/eLife.33569.009

To compare the transcription termination events in 3' UTRs and ORFs, we analyzed the nucleotide composition around the identified poly(A) sites. Similar to previous results from analysis of yeast transcription termination regions (*Tian and Graber, 2012*), the sequence profile surrounding 3' UTR-pA sites in *Neurospora* has an upstream A-rich region located at −30 to −10 nucleotides from poly(A) site and two U-rich regions at −10 to 0 nucleotides and at 0 to +10 nucleotides (*Figure 4C*). However, the U-rich region immediately downstream of the poly(A) site is much less prominent than that in yeast (see below). The poly(A) site is usually located between C and A nucleotides. A motif search of most enriched hexamers within the A-rich region was performed to identify putative PAS motifs (*Ulitsky et al., 2012*). In mammals, AAUAAA and AUUAAA are the two most frequently used PAS motifs and are found in ~80% of all 3' UTR-PAS (*Tian and Graber, 2012*; *Proudfoot, 2011*; *Manley, 2015*). The PAS motifs in *Neurospora* are much more degenerated with AAUGAA being the most abundant and AAUAAA is the third most-enriched motif (*Figure 4D*).

Although the nucleotide profile surrounding ORF-pA sites was similar to that of 3' UTR-pA sites with A-rich and U-rich elements upstream of the C/A poly(A) site, there does not appear to be a U-rich region downstream (*Figure 4E*). In addition, the hexamer motifs in the A-rich region of ORF-PASs were quite degenerative (*Figure 4F*). Among the top 15 most enriched hexamer motifs, only five were shared between ORF-PAS and 3' UTR-PAS regions (*Figure 4D and F*). To further compare 3' UTR-PAS and ORF-PAS, we generated consensus PAS sequences to build position-specific scoring matrices (PSSMs) for PAS regions by using sequences (−30 ~+10 nt) as previously described (*Tian et al., 2007*). The PSSMs were then used to score all 3' UTR-PASs and ORF-PASs. A high PAS score indicates a high similarity to the consensus and, presumably, a stronger signal for transcription termination. As shown in *Figure 4G*, ORF-PASs generally show lower PAS scores than that of 3' UTR-PASs. These results suggest that premature transcription termination within ORFs occurs through a mechanism similar to that in the 3' UTR with recognition of the poly(A) site mostly mediated by non-canonical poly(A) signals.

## Strong genome-wide correlations between codon usage and premature transcription termination

To understand the role of codon usage in PCPA, we examined the genome-wide correlations between gene codon usage and transcription termination events within *Neurospora* ORFs. Based on the 2P-seq results, we selected 2957 genes (RPM >10) that have ORF-pA sites and calculated the normalized ratio between the numbers of termination events in the ORFs and in the 3' UTRs. The ratios were less than 10% for 95% of the genes with ORF-pA, which should be due to that these non-canonical poly(A) signals within ORFs are less efficient in promoting premature cleavage and polyadenylation (*Berg et al., 2012*; *Guo et al., 2011*) or that the premature terminated RNAs are unstable (*van Hoof et al., 2002*; *Frischmeyer et al., 2002*; *Doma and Parker, 2007*; *Vanacova and Stefl, 2007*). We also calculated the codon bias index (CBI) and codon adaptation index (CAI) for every protein-coding gene in *Neurospora* (*Bennetzen and Hall, 1982*; *Sharp and Li, 1987*). The normalized values of ORF to 3' UTR termination events showed a strong negative correlation with both CBI and CAI (*Figure 5A and B*). These results suggest that codon usage, by affecting the formation of potential poly(A) signals, plays an important role in PCPA in *Neurospora*. For *Neurospora* genes, there is a strong preference for C/G at the wobble positions, thus genes with more rare codons should have higher AU contents and potentially higher chance of forming poly(A) signals to trigger premature termination.

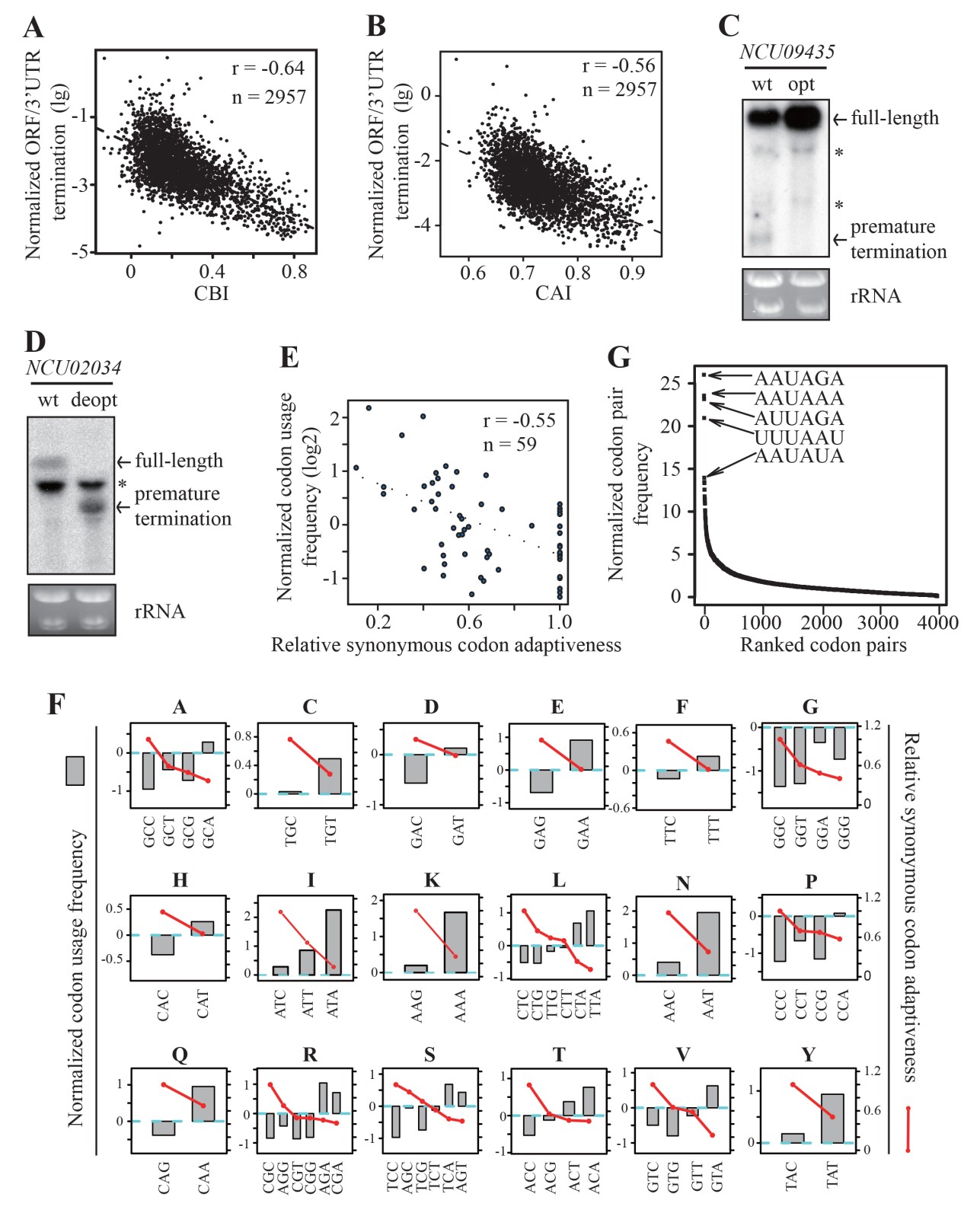

**Figure 5.** Strong genome-wide correlations between codon usage and premature transcription termination events. (**A**) Scatter plot of normalized ORF/ 3' UTR termination events (log10) vs. CBI. $r = -0.64$, $p < 2.2 \times 10^{-16}$, n = 2957. (**B**) Scatter analysis showing the correlation of normalized ORF/3' UTR termination events with CAI. Pearson's $r = -0.56$. $p < 2.2 \; 10^{-16}$, n = 2957. (**C**) Northern blot analyses showing that premature transcription termination was abolished after codon optimization of *NCU09435*. *gfp-NCU09435*-wt and *gfp-NCU09435*-opt were targeted to the *his-3* locus, and an RNA probe
*Figure 5 continued on next page*

*Figure 5 continued*

specific for *gfp* was used. The asterisks indicate non-specific bands. (D) Northern blot analyses showing that premature transcription termination was observed after codon de-optimization of *NCU02034*. *gfp-NCU02034*-wt and *gfp-NCU02034*-deopt were targeted to the *his-3* locus, and an RNA probe specific for *gfp* was used. (E) Scatter plot of normalized codon usage frequency (NCUF) (log2) with relative synonymous codon adaptiveness (RSCA) of all codons with at least two synonymous codons. *r* = −0.55, p=3.8 × 10⁻⁶, n = 59. (F) The correlation of normalized codon usage frequency (NCUF) with relative synonymous codon adaptiveness (RSCA) within each synonymous codon group with at least two synonymous codons. NCUF values of every codon within the −10 to −30 regions upstream of all identified ORF-pA sites was calculated. (G) A graph showing the ranking of all codon pairs by normalized codon pair frequency (NCPF). Codon pairs are ranked based on their NCPF values.

DOI: https://doi.org/10.7554/eLife.33569.010

The following figure supplements are available for figure 5:

**Figure supplement 1.** Sequence alignment of *NCU09435*-wt and *NCU09435*-opt construct s (A) and  sequence alignment of *NCU02034*-wt and *NCU02034*-deopt constructs (B).

DOI: https://doi.org/10.7554/eLife.33569.011

**Figure supplement 2.** Nucleotide composition (U top panel and A bottom panel) surrounding PAS motifs in 3' UTR (A) and coding region (B).

DOI: https://doi.org/10.7554/eLife.33569.012

**Figure supplement 3.** Correlations between relative codon usage frequency and relative synonymous codon adaptiveness.

DOI: https://doi.org/10.7554/eLife.33569.013

To confirm this conclusion, we tested whether a premature termination event within ORF can be abolished by optimizing surrounding rare codons or created by introducing rare codons within an ORF. *NCU09435,* which has a prominent termination site at the 5' end of its ORF (*Figure 4A*), was selected as a reporter gene. The *NCU09435* ORF was inserted downstream of *gfp* to create the *gfp-NCU09435*-wt construct. The 19 non-optimal codons surrounding the premature termination site of *NCU09435* were replaced by optimal codons to create the *gfp-NCU09435*-opt construct (*Figure 5—figure supplement 1A*). Codon optimization led to not only the loss of the putative PAS motif (AAU-CAA) but also mutation of its immedi    ate surrounding sequence. Northern blot analysis showed that the codon optimization completely abolished the premature termination product of the *gfp-NCU09435* fusion gene (*Figure 5C*), indicating the importance of the PAS motif and its surrounding sequence on premature transcription termination.

*NCU02034* has no detectable premature termination events even though it has a potential PAS motif (AAUAAA) within its ORF. The 16 optimal codons surrounding the potential PAS motif in the *gfp-NCU02034*-wt were replaced by rare synonymous codons to create the *gfp-NCU02034*-deopt construct (*Figure 5—figure supplement 1B*). Using this reporter, we showed that codon deoptimization abolished the expression of full-length transcript, and transcription was prematurely terminated around the deoptimized region (*Figure 5D*). These results demonstrate the importance of codon usage and *cis*-elements around the PAS motif in promoting premature transcription termination.

To further understand how codon usage bias influences premature transcription termination, we determined the codon usage frequency of every codon within the −10 to −30 regions upstream of all identified ORF-pA sites. The frequency of each codon was then normalized to the frequency of the same codon from randomly chosen regions from the same gene set to create the normalized codon usage frequency (NCUF). For a given codon, a positive NCUF value means that this codon occurs more frequently within the PAS region relative to other regions of the gene; a negative value indicates that a codon occurs less frequently. Overall, there is a strong negative correlation (r = −0.55) between NCUF with the relative synonymous codon adaptiveness (RSCA) (*Roth et al., 2012*) of all codon groups with at least two synonymous codons (*Figure 5E*). Moreover, as shown in *Figure 5F*, amino acids encoded by A/T-rich codons, such as isoleucine, lysine, asparagine, and tyrosine, are over-represented within the region upstream of poly(A) sites in ORFs. These results indicate that rare codons are preferentially used upstream of the poly(A) sites within ORFs.

In addition to codon usage bias, pairs of synonymous codons are also not used equally in the genome, a phenomenon called codon pair bias (*Gutman and Hatfield, 1989*; *Moura et al., 2007*). Since PAS motifs are hexamers, we examined codon pair usage in the region surrounding the ORF-pA sites. We determined codon pair frequencies in 12106 ORF-PAS sequences (−30 ~ −10nt) that belongs to 4557 genes and normalized the values with background codon pair frequencies randomly chosen from the same gene set to generate normalized codon pair frequency (NCPF). For a given

codon pair, a positive value of NCPF means that this codon pair is over-represented. Remarkably, when codon pairs are ranked based on NCPF, the five most enriched codon pairs are all made of two of the least preferred codons (*Figure 5G*). For example, AAUAGA is the most enriched codon pair and is 25-fold enriched in this region relative to the genome frequency and is one of the most often observed PAS motifs in 3' UTRs and ORFs (*Figure 4D and F*). These results support our hypothesis that the use of rare codons leads to the formation of potential PAS motifs and transcription termination, whereas the use of preferred codons in *Neurospora* genes suppresses premature transcription termination events to ensure expression of full-length mRNAs by depleting the formation of potential PAS motifs.

To examine whether the presence of single PAS motif is sufficient to terminate transcription, we performed the following analysis. We searched for all the DNA fragments containing a putative PAS motif in ORFs and 3' UTRs of all *Neurospora* genes and divided them into two groups: 'true PAS', which have at least one identified pA site within the 5-35nt downstream region, and 'false PAS', which don't have a pA site (see Materials and methods for details). In 3' UTR, 17404 are true PAS motifs and 20643 false PAS motifs. We compared the nucleotide composition between these two groups and found out that the AU contents of surrounding the true PAS motifs were higher than those of the false PAS motifs (*Figure 5—figure supplement 2A*). In the coding region, 4468 true PAS and 80086 false PAS were found. The ratio of true PAS to false PAS in ORF (true PAS/false PAS=0.086) is much lower than that in the 3' UTR (1.03), indicating that a single PAS motif alone is not sufficient to trigger PCPA in the ORFs. Similar to the 3' UTR, AU contents of surrounding true PAS motifs were also higher than those of false PAS motifs (*Figure 5—figure supplement 2B*). These results indicate that *cis*-elements surrounding PAS are also important for transcription termination in the 3' UTR and ORF. Consistent with this conclusion, a false PAS in *NCU02034* can be turned into a true PAS by deoptimizing the codons surrounding PAS motif (*Figure 5D*).

We then compared the codon usage surrounding true PAS or false PAS (see Materials and Methods). We determined the codon usage frequency of every codon surrounding true PAS and then normalized by that of the same codon surrounding false PAS to create the relative codon usage frequency (RCUF) (see Method and Material for details). As shown in *Figure 5—figure supplement 3A*, there was a strong negative correlation ($r = -0.66$) between RCUF with RSCA of all codon groups with at least two synonymous codons. Moreover, a negative correlation was found between RCUF and RSCA within every codon group (*Figure 5—figure supplement 3B*). These results indicate that rare codons are preferentially used in the regions surrounding true PAS motif and are critical for promoting premature transcription termination. Together, these results suggest that codon usage affects premature transcription termination in *Neurospora* by creating potential PAS motif and its surrounding *cis*-elements.

## Transcription termination events in *Schizosaccharomyces pombe*, an organism with A/U-biased codon usage

The mechanism of transcription termination is largely conserved in eukaryotic organisms and requires similar A/U-rich motifs despite distinct codon usage biases of different genomes. Our data indicate that codon usage bias influences transcription termination in *Neurospora*, which has a C/G-biased genome. To evaluate how A/U-biased organisms prevent frequent premature transcription termination, we evaluated data from the fission yeast *Schizosaccharomyces pombe*. *S. pombe* has a strong A/U bias for every codon family, and its gene expression levels are correlated with gene codon usage bias (*Hiraoka et al., 2009*; *Forsburg, 1994*). Using the high-quality poly(A)-seq data recently generated for *S. pombe* (*Lemay et al., 2016*), we identified 7429 ORF-pA sites in 2894 genes. As shown in *Figure 6A*, there is an A-rich region in the region −30 to −10 nucleotides upstream and two equally prominent U-rich regions flanking the poly(A) sites in the 3' UTR. The nucleotide composition of the poly(A) sites within ORFs has a very similar profile with a prominent downstream U-rich domain, which is missing in *Neurospora*. This result suggests that the downstream U-rich region is required for transcription termination, a conclusion that is supported by previous experimental studies (*Dichtl and Keller, 2001*; *Graber et al., 1999*; *van Helden et al., 2000*). Comparison of the poly(A) sites in 3' UTRs with those within ORFs showed that the ORF-pA sites have significantly lower PAS scores than those in 3' UTR (*Figure 6B*).

Compared to *S. pombe*, the U-rich region downstream of poly(A) sites in *Neurospora* 3' UTRs is much less prominent (*Figure 4C*), and the downstream U-rich region is absent for poly(A) sites within

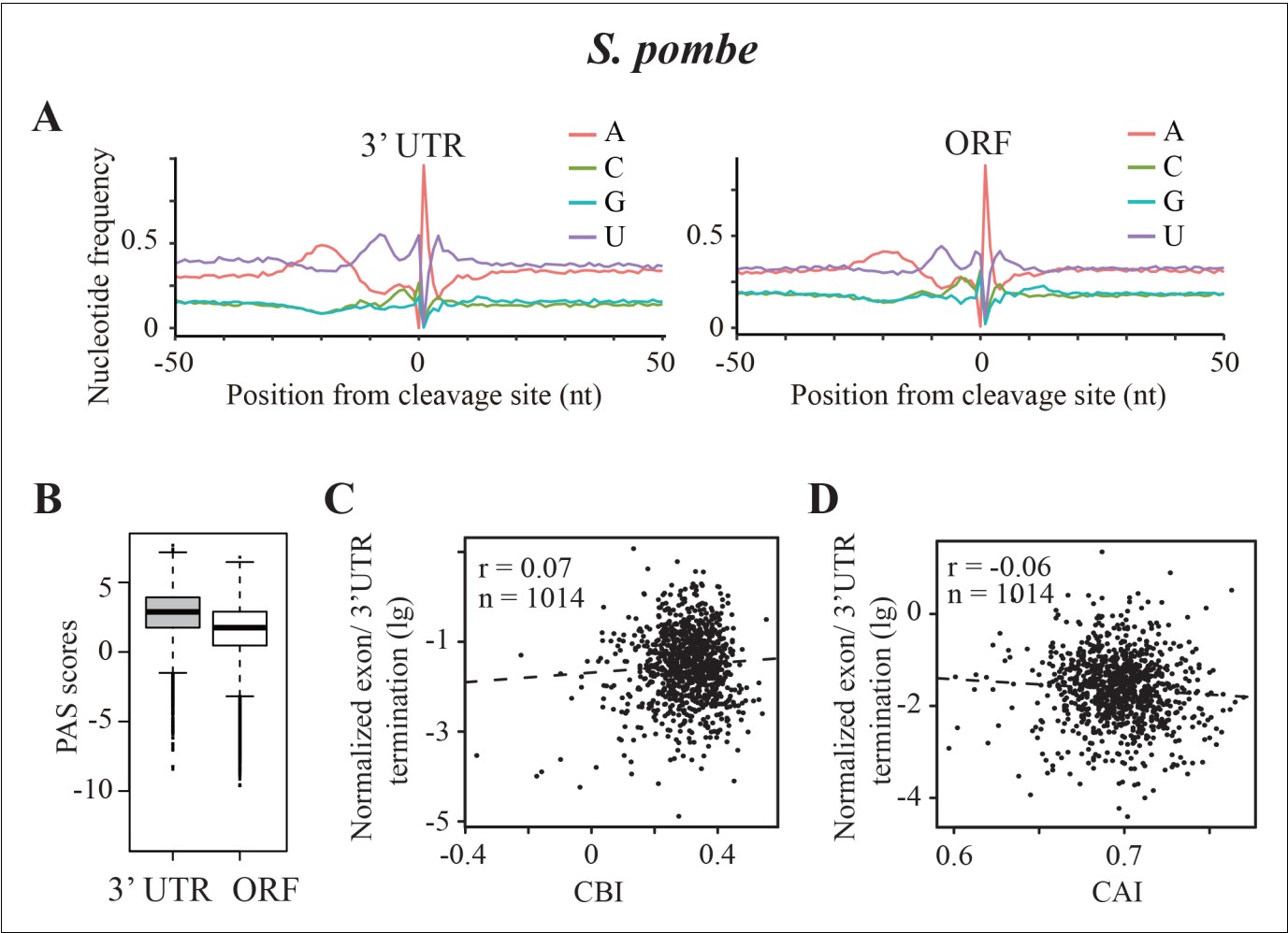

**Figure 6.** Transcription termination events in *Schizosaccharomyces pombe*. (**A**) Nucleotide sequence composition surrounding the poly(A) sites located in 3' UTR (left) and in ORF (right) in *S. pombe*. (**B**) Genome-wide PAS scores for 3'UTR-PAS and ORF-PAS. (**C**) Scatter analysis showing the correlation of normalized ORF/3' UTR termination with CBI. *r* = 0.07, p=0.04, n = 1014. (**D**) Scatter plot of normalized ORF/3' UTR termination vs CAI. *r* = −0.06, p=0.03, n = 1014.

DOI: https://doi.org/10.7554/eLife.33569.014

ORFs in *Neurospora* (*Figure 4E*). This suggests that the U-rich region just downstream of the poly(A) site is not an essential element for transcription termination in ORFs in *Neurospora*. Therefore, the nucleotide sequence requirement for transcription termination in ORFs in *S. pombe* appears to be more stringent than that in *Neurospora*. This provides an explanation for how an A/U-biased organism such as *S. pombe* can have highly expressed A/U-rich genes without prominent PCPA in ORF. Consistent with this notion, in *S. pombe* the normalized values of ORF to 3' UTR termination events have little or no correlation with CBI or CAI values (*Figure 6C and D*). Thus, organisms with C/G and A/U codon usage biases in fungi appear to use different mechanisms to adapt transcription termination process to allow optimal gene transcription.

## A conserved role for codon usage in transcription termination in mouse

As *Neurospora*, the codon usage of mammalian genomes is also C/G-biased. Although PAS signals in mammals are also AU-rich sequences, additional *cis*-elements, including downstream GU-rich and G-rich elements, are also required for efficient polyadenylation and transcription termination (*Tian and Graber, 2012*). In addition, protein factors or complexes involved in polyadenylation are only partially conserved between fungi and mammals (*Tian and Graber, 2012*; *Shi and Manley, 2015*). To examine whether premature transcription termination is also affected by codon usage

bias in mammals, we analyzed the recently available high-quality poly(A)-seq data from mouse (*Yang et al., 2016*). By combining the data from four replicates of C2C12 myoblast control samples, we identified 2564 poly(A) sites within the ORFs of 1429 genes. We compared the nucleotide profile of ORF-PAS to that of 3' UTR-PAS and we found that the upstream U-rich region (−15 to −1) and the downstream GU-rich region (+5 to+20) largely disappeared and the AU content decreased in ORF-PASs, suggesting that non-canonical poly(A) signals are used for the premature transcription termination in ORFs in mouse (*Figure 7A*). Consistent with this result, we found that ORF-PASs have significantly lower PAS scores than 3'UTR-PAS (*Figure 7B*). Moreover, although AAUAAA and AUUAAA are the top two most frequently used PAS motifs in ORF-PAS, their frequencies are much lower than to that in 3'UTR (*Figure 7—figure supplement 1A and B*). As in other organisms (*Tian et al., 2007*; *Liu et al., 2017b*), widespread premature transcription termination events in introns were found in mouse with 10923 poly(A) sites in the introns of 8345 genes. Different from ORF-PAS, however, the nucleotide profile of the intronic PAS was almost identical to that of 3' UTR-PAS (*Figure 7—figure supplement 1C*) and the frequency of AAUAAA and AUUAAA was only slightly lower than that in the 3' UTR (*Figure 7—figure supplement 1D*). Together, these data suggest that a similar mechanism as that of 3'UTR are employed for premature transcription termination in ORFs with the tendency to use non-canonical poly(A) signals.

We examined the correlations between gene codon usage and transcription termination events within mouse ORFs. We found that there was a negative correlation between normalized ORF/3'UTR termination and CBI (*Figure 7C*) or CAI (*Figure 7D*), suggesting that premature transcription termination is also affected by codon usage in mouse. Although C or G is also preferred at the wobble positions in mouse, its codon usage bias is not as strong as that is observed in *Neurospora* (*Zhou et al., 2015*). Moreover, unlike in *Neurospora*, ORFs in mouse are often consist of small exons, which are separated by large introns. These facts may explain the negative correlations between the normalized ORF/3'UTR termination and codon usage bias in mouse is not as strong as that in *Neurospora* (*Figure 5A and B*). Next, we calculated the normalized codon usage frequency for the −10 to −30 regions upstream of all ORF-pA sites in mouse. As in *Neurospora*, a negative correlation between NCUF and RSCA was observed (*Figure 7—figure supplement 2A*). Moreover, the negative correlation between NCUF and RSCA was found within most synonymous codon groups, except for histidine (*Figure 7E*). In addition, PAS motifs (AAUAAA and AUUAAA) were among the top three most enriched codon pairs and are over 16-fold enriched in this region relative to the genome frequency (*Figure 7—figure supplement 2B*). These results indicate that codon preference surrounding ORF-PAS is substantially different from the rest of ORFs. As in *Neurospora*, the nucleotide U content surrounding the true PAS motifs of 3' UTR and ORF was higher than that of the false PAS in mouse (*Figure 7—figure supplement 2C-D*). Together, these analyses suggest that codon usage biases co-evolved with transcription termination machinery to limit premature transcription termination in C/G-biased organisms from *Neurospora* to mouse.

## Discussion

Codon usage is an important determinant of protein expression levels in different organisms. The effects of codon usage bias were thought mainly due to its influence on protein translation. We previously showed that codon usage affects chromatin structures and is a major determinant of mRNA levels by affecting gene transcription (*Zhou et al., 2016*), indicating that codon usage can impact gene expression beyond translation. In this study, we demonstrated that premature cleavage and polyadenylation in coding region is affected by codon usage biases.

Our conclusion is supported by several lines of evidence. First, replacing wild-type codons with rare codon in parts of the circadian clock gene *frq* abolished expression of the full-length mRNA transcript due to premature transcription termination. Second, by identifying the rare codons important for premature transcription termination, we showed that rare codons created potential poly(A) signals, including PAS motif and other upstream and downstream *cis*-elements. Introduction of a single rare codon in the correct context caused premature transcription termination and had a significant effect on *frq* gene expression. In contrast, the use of optimal codons can suppress premature transcription termination and promote the expression of full-length transcript. Third, genome-wide identification of mRNA poly(A) sites uncovered 12106 premature termination sites in the ORFs of 4557 genes, indicating that premature transcription termination within ORFs is a wide-spread

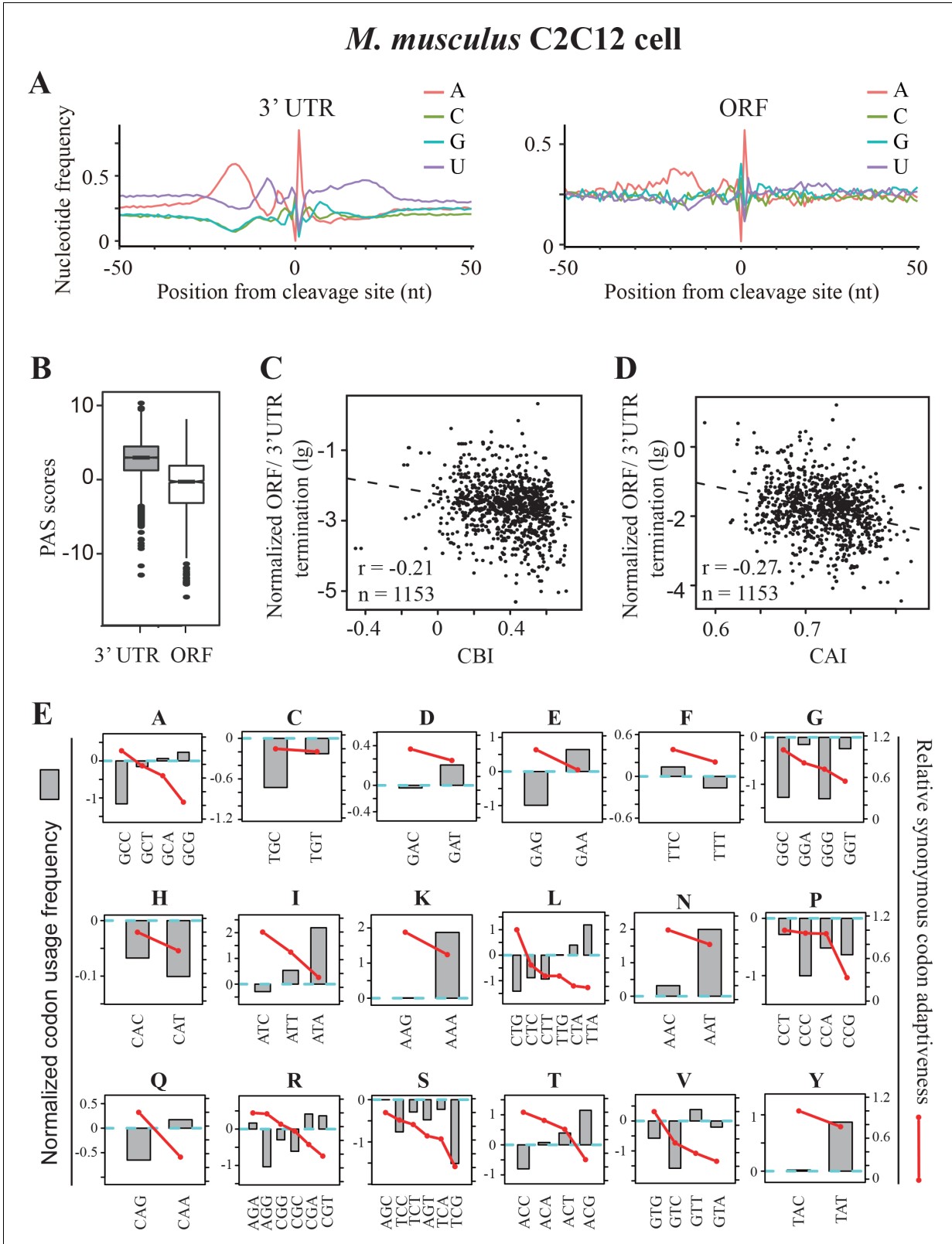

**Figure 7.** Premature transcription termination events in ORFs in mouse C2C12 cells. (**A**) Nucleotide sequence composition surrounding poly(A) sites in 3′ UTR (left) and in ORFs (right) in mouse C2C12 cells. (**B**) PAS scores for 3′UTR-PAS and ORF-PAS. (**C**) Scatter analysis showing the correlation of normalized ORF/3′ UTR termination with CBI. Pearson's $r = -0.21$. $p=6.29 \times 10^{-13}$, $n = 1153$. (**D**) Scatter plot of normalized ORF/3′ UTR termination vs CAI. $r = -0.27$, $p<2.2 \times 10^{-16}$, $n = 1153$. (**E**) The correlation of normalized codon usage frequency (NCUF) with relative synonymous codon

*Figure 7 continued on next page*

Figure 7 continued

adaptiveness (RSCA) within each synonymous codon group with at least two codons. NCUF values were calculated for −10 to −30 nt regions upstream of identified poly(A) sites in ORFs.

DOI: https://doi.org/10.7554/eLife.33569.015

The following figure supplements are available for figure 7:

**Figure supplement 1.** Sequence analyses of poly(A) sites in mouse C2C12 cells.

DOI: https://doi.org/10.7554/eLife.33569.016

**Figure supplement 2.** Codon usage and sequence analyses of poly(A) sites in mouse C2C12 cells.

DOI: https://doi.org/10.7554/eLife.33569.017

phenomenon in *Neurospora*. Importantly, we discovered a strong genome-wide negative correlation between gene codon usage and premature transcription termination events in *Neurospora* genes. Fourth, the regions around the premature termination sites are highly enriched in rare codons and rare codon pairs. Fifth, by comparing nucleotide profile and codon usage bias between true PAS and false PAS, we found that rare codons not only create PAS motif but also other surrounding *cis*-elements important for transcription termination. Since gene codon usage strongly correlates with mRNA and protein expression levels in *Neurospora* (*Zhou et al., 2016*), these results suggest that differential effects of synonymous codons on transcription and premature transcription termination contribute to the determination of gene expression levels. Consistent with our conclusion, a previous study in yeast has shown that codon usage affects stability of antisense transcripts by generation or depletion of the Nrd1 and Nab3 binding sites (*Cakiroglu et al., 2016*).

The codon usage of the *Neurospora* genome is strongly C/G biased at the wobble positions. Transcription termination in eukaryotes relies on multiple A/U rich *cis*-elements around the poly(A) sites (*Tian and Graber, 2012*; *Shi and Manley, 2015*; *Proudfoot, 2011*; *Proudfoot, 2016*; *Tian and Manley, 2017*; *Kuehner et al., 2011*). Clusters of rare codons in *Neurospora* appear to form the A/U rich poly(A) signal that can be recognized by transcription termination machinery. Our analyses of poly(A)-seq results in mouse cells, which also has a C/G-biased codon usage, suggest that codon usage bias also play a role in suppressing premature transcription termination within ORFs. Therefore, codon usage bias is a conserved mechanism for C/G-biased genomes to promote gene expression by suppressing PTT. Consistent with this conclusion, it has been previously shown that codon optimization is required for high-level expression of heterologous genes in *Aspergillus oryzae*, probably by preventing premature transcription termination in an exon (*Tokuoka et al., 2008*).

Although a similar mechanism as that of 3'UTR is employed, non-canonical poly(A) signals are used for premature cleavage and polyadenylation within coding regions. First, the nucleotide profile of ORF-PAS is different from that of 3' UTR-PAS. In *Neurospora,* the downstream U-rich region is missing (*Figure 4E*) whereas the upstream U-rich region and the downstream GU-rich region largely disappeared in mouse (*Figure 7A*). Second, compared to the PAS motifs in 3' UTR, the non-canonical poly(A) signals within ORFs are less efficient to terminate transcription, resulting in the low ratio of ORF to 3' UTR termination events observed both in *Neurospora* and mouse (*Figure 5A and B* and *Figure 7C and D*). The use of non-canonical poly(A) signals may also affect the fate of prematurely terminated transcripts. It has been shown that prematurely terminated transcripts are rapidly degraded in the cytosol and in the nucleus (*van Hoof et al., 2002*; *Frischmeyer et al., 2002*; *Doma and Parker, 2007*; *Vanacova and Stefl, 2007*). In *Neurospora* and mouse, the ratio of true PAS motifs in coding regions is much lower than that in the 3' UTR, suggesting that PCPA in coding region is mostly suppressed by optimal codons surrounding PAS signal. Even though the non-canonical poly(A) signals are less efficient, a gene with poor codon usage can have multiple such sites, which can have a major impact on mRNA levels.

By analyzing the mRNA termination sites in *S. pombe,* which has a strong A/U-biased codon usage, we also identified many premature transcription termination events in ORFs. In contrast to *Neurospora*, there is little or no correlation between codon usage and premature transcription termination events in *S. pombe,* suggesting that codon usage does not contribute significantly to premature transcription in organisms with A/U-biased genomes. Comparison of the nucleotide composition around transcription termination sites in 3' UTRs and in ORFs in *S. pombe* showed that they share a U-rich motif downstream of the cleavage sites, which has been shown to be important for transcription termination (*Dichtl and Keller, 2001*; *Graber et al., 1999*; *van Helden et al.,*

2000). Such a U-rich element is largely missing in *Neurospora*. These results suggest that both C/G- and A/U-biased genomes adapt with transcription termination mechanisms to use different mechanisms to prevent premature transcription termination. The C/G-biased organisms such as *Neurospora* use C/G-biased codons to prevent the formation of poly(A) signals, thus suppressing premature transcription termination, whereas the A/U-biased organisms relies on more stringent sequence requirements for poly(A) signals.

# Materials and methods

**Key resources table**

| Reagent type (species) or resource | Designation | Source or reference | Identifiers | Additional information |
|---|---|---|---|---|
| gene (*Neurospora crassa*) | *frequencey* (*frq*) | NA | NCBI Gene ID: 3876095 | |
| gene (*Neurospora crassa*) | *NCU09435* | NA | NCBI Gene ID: 3874734 | |
| gene (*Neurospora crassa*) | *NCU00931* | NA | NCBI Gene ID: 3880910 | |
| strain (*Neurospora crassa*) | 4200 | PMID:155773 | | Strain maintained in Yi Liu's lab |
| strain (*Neurospora crassa*) | 303–3 (*bd, frq10, his-3*) | PMID:8052643 | | |
| strain (*Neurospora crassa*) | 301–6 (*bd, his-3*, A) | PMCID: PMC180927 | | |
| antibody | anti-FRQ | PMID:9150146 | | Rabbit polyclonal; 1:50 for western blot |
| antibody | anti-WC-2 | PMID: 11226160 | | Rabbit polyclonal; 1:500 for ChIP |
| antibody | Anti-RNA polymerase II CTD repeat YSPTSPS (phospho S2) antibody | abcam | ab5095 | Rabbit polyclonal; 1:500 for ChIP |
| antibody | Anti-H3K9me3 | Active Motif | catalog no: 39161 | Rabbit polyclonal; 1:500 for ChIP |
| recombinant DNA reagent | pKAJ120 | PMID:8052643 | | deoptimized *frq* gene; see *Figure 1—figure supplement 1* |
| recombinant DNA reagent | *frq*-deopt1 | this paper | | deoptimized frq gene; see *Figure 1—figure supplement 1* |
| recombinant DNA reagent | *frq*-deopt2 | this paper | | deoptimized *frq* gene; see Materials and methods |
| recombinant DNA reagent | *frq*-deopt3 | this paper | | deoptimized *frq* gene; see Materials and methods |
| recombinant DNA reagent | frq-deopt4 | this paper | | deoptimized *frq* gene; see Materials and methods |
| recombinant DNA reagent | frq-deopt5 | this paper | | deoptimized *frq* gene; see Materials and methods |
| recombinant DNA reagent | frq-deopt6 | this paper | | deoptimized *frq* gene; see Materials and methods |
| recombinant DNA reagent | frq-deopt7 | this paper | | deoptimized *frq* gene; see Materials and methods |
| recombinant DNA reagent | frq-deopt4* | this paper | | deoptimized *frq* gene; see Materials and methods |
| recombinant DNA reagent | *gfp-NCU09435*-wt | this paper | | wild-type *NCU09435* gene in frame with *gfp* |
| recombinant DNA reagent | gfp-*NCU09435*-opt | this paper | | optimized *NCU09435* gene; see *Figure 5—figure supplement 1* |

*Continued on next page*

*Continued*

| Reagent type (species) or resource | Designation | Source or reference | Identifiers | Additional information |
|---|---|---|---|---|
| recombinant DNA reagent | gfp-NCU02034-wt | this paper | | wild-type NCU02034 gene in frame with gfp |
| commercial assay or kit | SuperScript III Reverse Transcriptase | Thermo Fisher (Waltham, MA ) | catalog no: 18080093 | For 3' RACE and making 2P-seq library |
| commercial assay or kit | TURBO DNA-free Kit | Thermo Fisher (Waltham, MA ) | catalog no: AM1907 | |
| commercial assay or kit | TOPO TA Cloning Kit, Dual promotor for in vitro Transcription | Thermo Fisher (Waltham, MA ) | catalog no: 452640 | |
| commercial assay or kit | Direct-zol RNA miniprep plus | Zymo research | catalog no: R2072 | |
| commercial assay or kit | CircLigase II ssDNA Ligase | Epicentre | catalog no: CL9021K | |
| software, algorithm | TopHat | http://ccb.jhu.edu/software/tophat/index.shtml | RRID:SCR_013035 | |
| software, algorithm | samtools | http://samtools.sourceforge.net/ | RRID:SCR_002105 | |
| software, algorithm | BEDTools | http://bedtools.readthedocs.io/en/latest/ | RRID:SCR_006646 | |
| software, algorithm | codonW | http://codonw.sourceforge.net/ | | |
| software, algorithm | Source code | this paper | | scripts to analyze 2P-seq and 3'READS. Including eight steps: read processing, mapping, filtering and downstream analyses that create plot and figures. |
| software, algorithm | raw sequencing data | this paper | PRJNA419320 | 2P-seq data, including two repeats from nuclear RNA extracts |
| software, algorithm | raw sequencing data | PMID:27401558 | GSE75753 | mouse poly(A)-seq data |
| software, algorithm | raw sequencing data | PMID:26765774 | GSE72574 | yeast poly(A)-seq data |

## Strains and culture conditions

In this study, FGSC 4200 (*a*) was used as the wild-type strain for 2P-seq. The 301–6 (*bd, his-3, A*) and 303–3 (*bd, frq*[10]*, his-3*) strains were the host strains for *his-3* targeting constructs. All the strains used in this study are listed in *Supplementary file 1*.

Culture conditions have been described previously (*Aronson et al., 1994b*). *Neurospora* mats were cut into small discs and transferred to flasks with minimal medium (1 × Vogel's, 2% glucose). After 24 hr, the tissues were harvested. Protein and RNA analyses were performed as previously described (*Zhou et al., 2016*). For race tube assay, the medium contains 1x Vogel's, 0.1% glucose, 0.17% arginine, 50 ng/ml biotin, and 1.5% agar. Strains were inoculated and grown in constant light at 25 degrees for 24 hr before being transferred to DD at 25 degrees. Growth fronts were marked every 24 hr. Calculations of period length were performed as described (*Garceau et al., 1997*).

## Codon deoptimization, plasmid constructs, and *Neurospora* transformation

*frq* codon deoptimization was performed for the 5' end of the ORF (36–489 nt). The nucleotide sequences of the deoptimized *frq* are shown in *Figure 1—figure supplement 1*. Sequences surrounding an alternative *frq* splice site in this region were not mutated. Codons were deoptimized based on the *N. crassa* codon usage frequency. In the *frq*-deopt1 construct, 65 codons were deoptimized, whereas 94 codons were changed in the *frq*-deopt2 construct. The deoptimized regions of *frq* were synthesized (Genscript) and inserted into *SphI* and *AflII* sites of pKAJ120 to generate *frq*-deopt1 and *frq*-deopt2. A homologous recombination-based cloning method (In-Fusion HD cloning kit, Clontech) was used to generate the *frq*-deopt3/4/5 constructs using *frq*-deopt2 as a template. In *frq*-deopt6, the 30[th] codon (GAC) of wild-type *frq* was mutated to GAT by site-directed mutagenesis. The 28[th] codon (TCC) of the *frq* ORF was mutated to AGT in *frq*-deopt7. AGT and GAT in the

*frq*-deopt4 construct were mutated to TCC and GAC, respectively, to make the *frq*-deopt4* construct.

The *qa-2* promoter inserted into pBM61 at *Not*I and *Xba*I sites to generate the pBM61.*qa-2* construct. The *gfp* ORF was inserted into pBM61.*qa-2* at *Xba*I and *Bam*HI sites to generate the pBM61.*qa-2-gfp* construct. The wild-type *NCU09435* ORF was inserted into pBM61.*qa-2-gfp* at *Bam*HI and *Hin*dIII sites to generate pBM61.*qa-2-gfp-NCU09435*-wt. The optimized region of *NCU09435* was synthesized (GenScript) and used to replace the corresponding region of the pBM61.*qa-2-gfp-NCU09435*-wt to generate pBM61.*qa-2-gfp-NCU09435*-opt. Wild-type *NCU02034* ORF was inserted into pBM61.*qa-2-gfp* at *Bam*HI and *Apa*I sites to generate pBM61. *qa-2-gfp-NCU02034*-wt. The deoptimized region of *NCU02034* was synthesized (GenScript) and used to replace the corresponding region of the pBM61.*qa-2-gfp-NCU02034*-wt to create pBM61.*qa-2-gfp-NCU02034*-deopt. All resulting *Neurospora* expression constructs were transformed into the host strains by electroporation as described previously (*Bell-Pedersen et al., 1996*). Homokaryotic transformants were obtained by microconidia purification and confirmed by PCR.

## Protein analyses

Tissue harvest, protein extraction, and western blot analyses were performed as previously described (*Garceau et al., 1997*; *Zhou et al., 2012*). For western blots, equal amounts of total protein (50 µg) were loaded in each lane. After electrophoresis, proteins were transferred onto PVDF membrane, and western blot analysis was performed. Anti-FRQ antibody, which was generated by using the full-length FRQ protein as antigen (*Garceau et al., 1997*), was used to detect FRQ protein levels in this study. For western blots, densitometry analyses were performed using Image J. To accurately quantify protein levels in different strains, a serial dilution was performed when needed.

## RNA, strand-specific qRT-PCR, northern blot, and 3' RACE

RNA extraction, strand-specific qRT-PCR, and northern blot were performed as previously described (*Xue et al., 2014*). Total RNA was extracted using Trizol and then purified with 2.5 M LiCl. Nuclear RNAs were isolated as described previously (*Zhou et al., 2016*). Briefly, nuclei were isolated, and nuclear RNAs were extracted using Direct-zol RNA miniprep plus kit (Zymo Research) according to the manufacturer's instruction.

Northern blot analyses were performed as previously described using [$^{32}$P] UTP-labeled riboprobes (*Xue et al., 2014*). Riboprobes were transcribed in vitro from PCR products by T3 or T7 RNA polymerase (Ambion) following the manufacturer's protocol. The primer sequences used for the template amplification were *frq*-5' end (5'-TAATACGACTCACTATAGGG (T7 promoter) GGCAGGG TTACGATTGGATT-3', 5'-GGGTAGTCGTGTACTTTGTCAG-3'), *frq*-3' end (5'-TAATACGACTCACTA TAGGG (T7 promoter) CCTTCGTTGGATATCCATCATG-3', 5'-GAATTCTTGCAGGGAAGCCGG-3'), *qrf*-5' end (5'-AATTAACCCTCACTAAAGGG (T3 promoter) GAATTCTTGCAGGGAAGCCGG-3', 5'-CCTTCGTTGGATATCCATCATG-3'), *gfp* (5'-TAATACGACTCACTATAGGG (T7 promoter) GAAC TCCAGCAGGACCATGTG-3', 5'-GAACCGCATCGAGCTGAA-3'), *NCU09435* 5' end (5'-TAATAC-GACTCACTATAGGG (T7 promoter) GAGGCAGCGTTAATGTTTGTG-3', 5'-GTGTCCAGTCAACTGG TTATCA-3'), and *NCU00931* 5' end (5'-TAATACGACTCACTATAGGG (T7 promoter) GGCAAGCGCCTTAAATTCTC-3', 5'-TACTCCCTGTCTTCAGTTCCT-3'). For northern blots, densitometry analyses were performed using Image J.

Strand-specific qRT-PCR was performed as previously described (*Xue et al., 2014*). cDNA was obtained by reverse transcription using a SuperScript III First-Strand Synthesis System (Invitrogen) using the manufacturer's instructions. *β-tubulin* was used for internal control. The primer sequences for strand-specific RT reactions were *frq* (5'GCTAGCTTCAGCTAGGCATC (adaptor) CGTTGCC TCCAACTCACGTTTCTT-3'), *frq* pre-mRNA (5'-GCTAGCTTCAGCTAGGCATC (adaptor) TTGAACGGTAGGGAGGAGGAGAG-3'), and *β-tubulin* (5'CTCGTTGTCAATGCAGAAGGTC-3'). The RT reaction was performed by mixing the primers of specific gene and *β-tubulin*. The primer sequences for the qPCR step of RT-qPCR assay were *frq* (5'-AGCTTCAGCTAGGCATCCGTT-3', 5'-GCAG TTTGGTTCCGACGTGATG-3'), *frq* pre-mRNA (5'-AGCTTCAGCTAGGCATCTTGAACG-3', 5'-ACGGCATCTCATCCATTCTCACCA-3'), and *β-tubulin* (5'-ATAACTTCGTCTTCGGCCAG-3', 5'-ACA TCGAGAACCTGGTCAAC-3').

3′ RACE was performed as previously described (*Scotto-Lavino et al., 2006*). Briefly, 2 µg total RNA was reverse transcribed using primer Qt (5′-CCAGTGAGCAGAGTGACGAGGACTCGAGC TCAAGCTTTTTTTTTTTTTTTTTTTT-3′). The 3′ end is amplified using a common primer $Q_o$ (5′-CCAG TGAGCAGAGTGACG-3′) and a gene specific primer GSP1 (*frq*, 5′-CCAACTCAAGTGCGTAAGGA-3′; *qrf*, 5′-GTCTTTCTCCTCTGCGATGTC-3′). The amplification product was diluted 40 times and was used as a template for the second amplification. Another common primer $Q_l$ (5′-GAGGAC TCGAGCTCAAGC-3′) and an inner gene-specific primer GSP2 (*frq*, 5′-CATGGCGGATAGTGGGGA TAA-3′; *qrf*, 5′-GGTAAGCCATGTACCTTGATCT-3′) were used for the second round of amplification. The second-round amplification product was cloned into TA cloning constructs (Invitrogen) and sequenced.

## 2P-seq

2P-seq was performed as described before (*Spies et al., 2013*) with several modifications. First, nuclear RNA was used instead of total RNA. Second, the primer sequence was re-designed (*Supplementary file 2*), so that multiple libraries could be sequenced in the same lane using Illumina sequencer. Third, a lower concentration of RT primer was used during reverse transcription to reduce internal priming (*Scotto-Lavino et al., 2006*). Briefly, poly(A) RNA was purified using oligo-dT25 beads (Invitrogen) and eluted directly into 25 µl of RNase T1 buffer. RNase T1 digestion was performed for 20 min at 22°C using 0.5 U RNase T1 (biochemistry grade, Ambion). After this partial RNase T1 digestion, reverse transcription was performed by addition of 1 µl 1 µM RT primer. Single-stranded cDNAs of 200–400 nucleotides were purified on a 6% TBE-urea gel, then circularized by CircLigaseII (Epicentre). Circularized cDNA was amplified by high-fidelity PCR for 10 cycles using barcoded PCR primers, then gel purified and sequenced using the Illumina Read one primer.

## Chromatin immunoprecipitation assay

ChIP assays were performed as described previously with some modifications (*Zhou et al., 2013b*). The tissues were cultured in 50 mL minimal medium and were fixed by adding 1% formaldehyde for 15 min at room temperature. The tissues were harvested and ground in the liquid nitrogen, and 100–200 µl tissue powder was suspended in 300 µl ChIP lysis buffer. Chromatin was sheared to about 500 bp fragments by sonication. In each reaction, 500 µg of total lysate and 2 µl antibody were added and incubated at 4°C overnight. The antibodies used in this study were: WC-2 (96), pol II S2P (Abcam, ab5095), and H3K9me3 (Active Motif, 39161). The 'Input' was 50 µg of total lysate. G-protein coupled beads (25 µl) were added, and samples were incubated for 2 hr. The beads washed at 4°C for 5 min with the following buffers: ChIP lysis buffer, low salt buffer, high salt buffer, LNDET buffer, and twice with TE buffer. We then added 140 µl 10% chelex beads (Sigma) and heated the samples at 96°C for 20 min. After centrifugation, 100 µl supernatant was transferred to a new tube. The Input DNA was de-crosslinked and extracted with phenol. Immunoprecipitated DNA was quantified using real-time quantitative PCR. For S2P and H3K9me3, the results were normalized to Input DNA and presented as Input %. For WC-2 ChIP, the results were further normalized to the internal control *β-tubulin*, and data are presented as relative WC-2 levels. Each experiment was performed independently three times. Primers used in the qPCR step are listed in *Supplementary file 2*.

## Data analyses

The poly(A)-seq data analyses were carried out with the combination of published tools and customized scripts written in Perl and R. All the scripts were available for download at Github (*Dang, 2018*; copy archived at https://github.com/elifesciences-publications/poly-A-seq).

## 2P-seq raw read processing for *N. crassa*:

Only raw reads with no fewer than 10 consecutive adenosines were used. Next, these poly(A) sequences and the adaptor sequences were removed, and any sequences less than 20 nt were discarded. Trimmed raw reads were mapped strand specifically to the *Neurospora crassa* genome (v10) using Tophat software (v2.1.1). The reads with multiple targets identified using SAMtools were discarded (*Li et al., 2009*). To identify reads due to false priming at internal poly(A) stretches of mRNA, we searched for genomic sequences in the 20 nt downstream of the 3′ end position of mapped

reads, and discarded the reads with six consecutive adenosines or ≥7 adenosines in 12-nt sliding windows (*Beaudoing et al., 2000*; *Tian et al., 2005*). These data were processed with BEDTools (*Quinlan and Hall, 2010*) (genomecov) to generate two bedgraph files, which reflect the density of reads at each position at plus or minus strand. The 2P-seq data were deposited in BioProject (accession# PRJNA419320)

### Raw read processing for *S. pombe* and mouse:

Fission yeast (GSE75753) (*Lemay et al., 2016*) and mouse (GSE72574) (*Yang et al., 2016*) poly(A)-seq data were previously generated. The raw read processing was performed as previously described (*Jan et al., 2011*). We performed the analysis using the following steps: 1) Remove adapter sequences and retain reads with at least one adenosine at the 3' end. 2) Remove the poly (A) stretch (≥1A), record the length of the poly(A) stretch for each read and discard the reads with a length shorter than 20 nt. 3) Map the processed reads with Tophat (v2.1.1) to the corresponding reference genome (Ensembl EF2 for *S. pombe* and UCSC mouse mm10, downloaded from iGenomes (Illumina). 4) Remove reads with multiple targets. 5) In order to avoid false positive signals, only the genomic sequences downstream of the 3' end of mapped reads and retained reads with at least one untemplated A were analyzed. 6) Create bedgraph files.

### Poly(A) site or pA site identification in 3' UTR and ORFs:

We first clustered sequences within 24 nt of the poly(A) site signals into peaks with BEDTools and recorded the number of reads falling in each peak (command: bedtools merge -s -d 24 c 4 -o count). We only retained those peaks with at least five reads for further analysis. We next determined the summit of each peak (i.e., the position with the highest signal) and took this peak to be the poly(A) site.

We classified the peaks into two different groups: peaks in 3' UTRs and peaks in ORFs. Because of the likely inaccurate 3' UTR annotations of genomic reference (i.e., GTF files of respective species), we set the 3' UTR regions of each gene from the end of the ORF to the annotated 3' end plus a 1-kbp extension. For a given gene, we analyzed all the peaks within the 3' UTR region, compared the summits of each peak and selected the position with the highest summit as the major poly(A) site of the gene.

For ORFs, we retained the putative poly(A) sites for which the PAS region fully overlapped with exons that are annotated as ORFs. The range of PAS regions for different species was empirically determined as a region with high AT content around the ORF poly(A) site. For each species, we did the first round of test setting the PAS region from −30 to −10 upstream of the cleavage site, then analyzed AT distributions around the cleavage sites in ORFs to identify the actual PAS region. The final settings for ORF PAS regions of *N. crassa* and mouse were −30 to −10 nt and those for *S. pombe* were −25 to −12 nt.

### Identification of 6-nucleotide PAS motif:

We followed the methods as previously described to identify PAS motifs (*Spies et al., 2013*). Specifically, we focused on the putative PAS regions from either 3' UTRs or ORFs. (1) We identified the most frequently occurring hexamer within PAS regions. (2) We calculated the dinucleotide frequencies of PAS regions, randomly shuffled the dinucleotides to create 1000 sequences, then counted the occurrence of the hexamer from step 1. (3) We tested the frequency of the hexamer from step one and retain it if its occurrence was ≥2 fold higher than that from random sequences (step 2) and if P-values were <0.05 (binomial probability). (4) We then removed all the PAS sequences containing the hexamer. We repeated steps 1 to 4 until the occurrence of the most common hexamer was <1% in the remaining sequences.

### Calculation of the normalized codon usage frequency (NCUF) in PAS regions within ORFs:

To calculate NCUF for codons and codon pairs, we did the following: For a given gene with poly(A) sites within ORF, we first extracted the nucleotide sequences of PAS regions that matched annotated codons (e.g., 6 codons within −30 to −10 upstream of ORF poly(A) site for *N. crassa*) and counted all codons and all possible codon pairs. We also randomly selected 10 sequences with the

same number of codons from the same ORFs and counted all possible codon and codon pairs. We repeated these steps for all genes with PAS signals in ORFs. We then normalized the frequency of each codon or codon pair from the ORF PAS regions to that from random regions.

### Relative synonymous codon adaptiveness (RSCA):

We first count all codons from all ORFs in a given genome. For a given codon, its RSCA value was calculated by dividing the number a particular codon with the most abundant synonymous codon. Therefore, for synonymous codons coding a given amino acid, the most abundant codons will have RSCA values as 1.

### Calculation of codon bias index (CBI):

The ORF sequences from *N. crassa* and *S. pombe* were extracted based on the genomic reference sequences. CBI for each gene was calculated with CodonW software (http://codonw.sourceforge.net/). Since condonW does not support mouse, CBI for each mouse gene is calculated as described (*Bennetzen and Hall, 1982*): CBI = $(N_{opt} - N_{ran})/(N_{tot} - N_{ran})$. $N_{opt}$, $N_{ran}$, and $N_{tot}$ represent the number of optimal, random and total codons in a given gene, respectively. We arbitrarily define optimal and random codons as RSCA $\geq 0.9$ or $0.3 <$ RSCA $< 0.9$, whereas codons with RSCA $<0.3$ are defined as rare codons.

### Calculation of codon adaptation index (CAI):

We calculated CAI values for all predicted protein-encoding genes in *N. crassa*, *S. pombe*, and mouse. We first calculated the frequencies of each codon in all annotated ORF sequences. For each amino acid, the relative synonymous codon adaptiveness (RSCA) for each synonymous codon was weighted by the most frequent synonymous codon. For each gene, CAI is defined as the geometric mean of RSCA values of all codons from the ORFs.

### Calculation of normalized ORF/3'UTR termination ratio:

For a given gene, the normalized ORF/3'UTR termination ratio reflects the relative frequencies of premature transcription termination events in ORFs. A higher value means that a gene has a higher chance to terminate transcription in ORF region. The ratio is calculated as below:

Normalized ratio = $N_{ORF-pA}*1000/ (N_{3' UTR-pA} *L)$

$N_{ORF-pA}$ and $N_{3' UTR-pA}$ stand for the number of reads (or pA events) in ORF and 3' UTR. L stands for the length of ORF.

### Calculation of PAS score:

The PAS score is calculated as previously described (*Tian et al., 2007*). Specifically, we extracted all sequences surrounding 3' UTR poly(A) sites (e.g. $-30$ ~+10 nt for *Neurospora*) to generate position-specific scoring matrices (PSSM). Each entry in the PSSM was calculated by $M_{ij} = \log_2 (f_{ij}/g_i)$, where $f_{ij}$ is the frequency of nucleotide i at position j, and $g_i$ is the genomic frequency of nucleotide i. For each PAS sequence, the PAS score is the sum of scores retrieved based on the position of nucleotides. It can be shown as S $=\sum_j m_{i, j}$, where $m_{i, j}$ is the score of nucleotide i at position j in the PSSM.

## Identification of sequence and codon context surrounding 6nt PAS motif

We selected the top 40 6nt PAS motifs in 3'UTR in *Neurospora*, which account for 58% of 3' UTR pA events of all expressed genes. For mice, we only selected the first 2 PAS motifs (AAUAAA and AUUAAA), which account for 77% of 3' UTR pA events. We first determined the genome-wide location of those motifs in a strand-specific manner. Then we classified these motifs into two major groups (true PAS and false PAS) based on whether the downstream region (5-35nt) after the motifs have pA signals at same strand. Each of these two groups is further categorized into two subgroups, based on whether these motifs are fully located inside the coding region (ORF true/false PAS) or at 3' UTR (UTR true/false PAS). For each group, 80nt DNA sequences flanking the motifs were collected and the A/T contents were calculated. Based on the A/T content difference between ORF true and false PAS group, we extracted codon sequences flanking the PAS motif with length equivalent. For *Neurospora*, we extract 15 codons at both upstream and downstream of PAS motif. For

each group, we first counted the occurrence of each codon and then divided it by the number of all codons from selected regions to obtain the ratio of each codon. For each codon, the relative codon frequency is the base two logarithm (log2) of the normalized ratio by dividing the ratio from ORF-true PAS group with that from ORF-false PAS group.

## Acknowledgements

We thank Dr. Noah Spies for providing the 2P-seq protocol and the members of our laboratory for technical assistance and discussion. This work is supported by grants from the National Institutes of Health (R35GM118118), Cancer Prevention and Research Institute of Texas (RP160268), and the Welch Foundation (I-1560) to Yi Liu.

## Additional information

### Funding

| Funder | Grant reference number | Author |
| --- | --- | --- |
| National Institute of General Medical Sciences | R35GM118118 | Yi Liu |
| Cancer Prevention and Research Institute of Texas | RP160268 | Yi Liu |
| Welch Foundation | I-1560 | Yi Liu |

The funders had no role in study design, data collection and interpretation, or the decision to submit the work for publication.

### Author contributions

Zhipeng Zhou, Conceptualization, Resources, Formal analysis, Validation, Investigation, Methodology, Writing—original draft, Writing—review and editing, Designed the study, performed almost all the Neurospora-related experiments, performed statistical analysis, interpreted the results; Yunkun Dang, Conceptualization, Resources, Data curation, Software, Formal analysis, Investigation, Methodology, Writing—original draft, Writing—review and editing, performed bioinformatic and statistical analyses, interpreted the results; Mian Zhou, Formal analysis, Investigation, Methodology, generated the frq-deopt1 and frq-deopt2 strains, interpreted the results; Haiyan Yuan, Resources, Investigation, contributed reagents, interpreted the results; Yi Liu, Conceptualization, Formal analysis, Supervision, Funding acquisition, Writing—original draft, Writing—review and editing, Designed the study, interpreted the results

### Author ORCIDs

Zhipeng Zhou (ID) http://orcid.org/0000-0002-8449-7194
Yi Liu (ID) http://orcid.org/0000-0002-8801-9317

### Decision letter and Author response

Decision letter https://doi.org/10.7554/eLife.33569.024
Author response https://doi.org/10.7554/eLife.33569.025

## Additional files

### Supplementary files

• Supplementary file 1. Strain list used in this study
DOI: https://doi.org/10.7554/eLife.33569.018

• Supplementary file 2. Primers used in this study
DOI: https://doi.org/10.7554/eLife.33569.019

• Transparent reporting form
DOI: https://doi.org/10.7554/eLife.33569.020

## Major datasets

The following dataset was generated:

| Author(s) | Year | Dataset title | Dataset URL | Database, license, and accessibility information |
|---|---|---|---|---|
| Zhou Z | 2017 | PolyA-seq | https://www.ncbi.nlm.nih.gov/bioproject/?term=PRJNA419320 | Publicly available at the NCBI BioProject database (Accession no: PRJNA419320) |

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
