## [Decision Letter]

Thank you for submitting your article "Codon usage biases suppress premature transcription termination to promote gene expression" for consideration by *eLife*. Your article has been favorably evaluated by Patricia Wittkopp (Senior Editor) and three reviewers, one of whom is a member of our Board of Reviewing Editors. The following individual involved in review of your submission has agreed to reveal her identity: Judith B Zaugg (Reviewer #3).

The reviewers have discussed the reviews with one another and the Reviewing Editor has drafted this decision to help you prepare a revised submission.

The presented data are of good quality and the observations are interesting. Yet, concerns are raised, of both analytical and interpretative character that should be addressed before publication in *eLife* can be considered. The major points below are essential to consider and we strongly suggest that the authors also improve the manuscript based on the additional considerations listed in the individual reviews.

Major points:

1) The title of the article is "Codon usage biases suppress premature transcription termination to promote gene expression". As formulated, it implies that there is actually an active mechanism that "senses" rare codons and feeds back this information to the transcription termination machinery. This would be very intriguing, but it appears that the most likely explanation is that codon bias on the one hand and transcription termination signal on the other hand co-evolved to adapt to a G/C rich genome. In other words, there is no direct cause and effect relationship between codon bias and transcription termination. Despite this likely explanation, the authors, in many instances, strongly indicate that there is a direct mechanism that couples the two phenomena. For example: "These results suggest that codon usage plays an important role in mediating premature transcription termination[…]"; "[…]suggesting that codon usage bias also plays a role in premature transcription termination in mouse"; "[…]codon usage biases are a conserved mechanism that affects premature transcription termination[…]"; "[…]codon usage bias is a conserved mechanism[…]".

These formulations are way too strong. Codon bias and transcription termination signals are correlated but it is not a "mechanism". Thus, these sentences, as well as the title, are misleading (obviously provocative, but also simply wrong) and the title should be changed and the text re-written accordingly.

2) A huge body of literature exists on premature transcription termination, which is ignored in the present manuscript. The findings should be discussed in the context of the relevant literature on (1) the link between codon bias and termination in *S. cerevisiae* and (2) premature termination at – typically intronic – premature cleavage and polyadenylation (PCPA) sites in mammals.

3) Since these rare codons can make up a canonical polyA site, more analyses on codon pairs would be welcome, and if it holds true that premature termination is not simply due to the formation of a PAS by two rare codons, it would be great if the authors could speculate on any other mechanism, e.g. by genome-wide analysis of ChIP-seq data. If it is not only the formation of a PAS (based on pair-wise analysis), what are the authors proposing as a mechanism? Is there any ChIP-seq data or similar available in Neurospora that shows an enrichment over the rare codons? For sure there is lots of ChIP-seq data available in mouse where the authors could look for factors potentially enriched at rare codons. Or does this coincide with PolII pausing? Moreover, the authors have excluded an effect on chromatin based on looking at one mark in one locus. Is there any genome-wide data available that the authors could use for checking this statement across the naturally occurring rare codons? For sure in mouse there would be enough data for that.

*Reviewer #1:*

This paper by Zhou et al. reports evidence for a role of codon usage biases affecting premature transcription termination in *Neurospora crassa* and *Mus musculus*. This was discovered through codon deoptimisation of the Neurospora *frq* gene, which leads to the loss of protein the full-length mRNA. The authors further show that this phenotype is due to premature transcription termination that generates short polyadenylated transcripts. Through generation of other mutants of the *frq* gene, Zhou et al. determined this mechanism to depend on multiple sequence elements that they further describe to be low-efficiency non-canonical transcription termination sites, based on genome wide analysis of endogenous 3'UTR and ORF polyadenylation sites. Characterisation of these polyadenylation sites revealed a genome wide correlation between codon usage and premature transcription termination. Finally, a similar genome wide analysis in *Saccharomyces pombe* and *Mus musculus* provided information suggesting that such a mechanism is not conserved in A/U-rich organism as *S. pombe* but might be present in G/C-rich ones as well (*M. musculus*).

Overall, the data shown are convincing and provide a conceptually new, though possibly not very surprising, link between codon bias and polyadenylation-dependent termination. The major caveat is that the findings are not discussed in the context of the relevant literature on (1) the link between codon bias and termination in *S. cerevisiae* and (2) premature termination at – typically intronic – premature cleavage and polyadenylation (PCPA) sites in mammals.

Additional points:

- (regarding deopt3): "suggesting that these products may be rapidly degraded" and "Because premature transcription termination products are usually rapidly degraded in the cytosol (van Hoof et al., 2002; Frischmeyer et al., 2002)". This point is probably of general relevance and should be discussed more explicitly. That is, the mostly low abundance of ORF PAS usage may well be due to the unstable nature of prematurely terminated transcripts. It is also worth noting that the prematurely terminated transcripts could well be subjected to nuclear decay as implicated in decay of human PCPA terminated RNAs, in addition to the non-stop decay suggested by the authors. This should be referenced together with a more general discussion of yeast and human premature termination systems.

- "The ratios were less than 10% for 95% of the genes with ORF-pA, indicating that only a small portion of transcription terminated in ORFs." The authors should discuss this more carefully. The RNA 3' ends do not mark transcription termination as RNA polymerase is present well beyond the PAS, as also found by the authors themselves. In addition, the low levels of ORF-pA may be partially due to the unstable nature of prematurely terminated transcripts.

- Figure 2: Is this understood correct that the authors claim that the antisense gene is also prematurely terminated when the sense *frq* is codon deoptimized? This warrants further explanation. Where does the premature termination occur and do the mutation create PAS on the antisense? I wonder whether this is some kind of artefact. If no plausible explanation for the premature termination of the antisense gene can be found also this should be mentioned.

- The authors find that ORF PAS are generally "weaker". Could this be a technical issue since those sites are generally also less abundant and hence less well defined? On a different note, the experimentally found pA events may be derived from alternative PAS-independent events. This could be discussed.

- In the case of *Mus musculus* data, it may be interesting to further compare to PAS motifs of intronic PCPA sites.

- Figure 5: The authors show that the AAUAAA element in *NCU02034* is not used due to optimal codons in the vicinity. This may allow for a supporting bioinformatic analysis where the codon optimality surrounding all ORF AAUAAA (or perhaps the AAUGAA) motifs is compared for genes, which use those sites, compared to genes where they are not used.

- Figure 1: despite the confirmation of the absence of protein by the absence of mRNA (Figure 1), some information regarding the anti-FRQ antibody would be needed (these do not seem to be provided either in the previous references). Is this antibody homemade? Which part of the protein does it target?

- Figure 3 proper Western blot quantification would require a serial dilution to estimate the protein concentration range in which the signal is linear (such a dilution do not require to be shown but should be mentioned in the Materials and methods).

- "other *cis*-elements such as U-rich upstream auxiliary element and U-rich downstream element also play important roles. In plants and yeast, the downstream element can be replaced by a U-rich element." These sentences are confusing and should be rephrased.

- Potential typo: "-30 ~ -10".

- Discussion concerning the previous findings that codon usage affects chromatin structure and transcription initiation. For the reader it would be interesting to relate those earlier findings in the context of the premature termination. That is, is it possible that changes in chromatin structure and transcription levels are indirectly due to premature termination and vice versa?

*Reviewer #2:*

In this manuscript, the authors report interesting observations. *Neurospora crassa* exhibits a strong GC rich sequence bias, which translates in a strong codon bias with synonymous codons with G or C at the wobble position being preferred. When changing codons of the *frq* gene to non-optimal codons, which are thus overall more A/T rich, they observed a complete disappearance of not only the expressed FRQ protein but also of the full-length mRNA.

They further show that it results from a different mechanism than the one described by the same group (Zhou et al., 2016), where they showed that a poor codon bias in *N. crassa* results in a poor transcription efficiency of the gene (and not a decreased stability as one would have expected). Here they show that the effect results from a premature termination event with a short transcript accumulating in place of the mRNA.

Performing the genome-wide mapping of mRNA 3'-ends, they identify a number of poly(A) sites (PAS) within ORFs and show that the occurrence of these are strongly correlated with a poor codon bias, genes enriched with rare codons being much more prone to exhibit transcription termination within ORFs. They show that this is not true for the A/T rich *S. pombe* but also true, yet to a much lower extent, for the mouse, which also exhibits a G/C enriched genome.

They conclude, "[…]codon biases are conserved mechanisms that affects premature transcription termination events in C/G-biased organisms[…]".

The presented data are of quality and the observations undoubtedly interesting.

Yet, I have some strong concerns that, I think, should be addressed before considering publication in *eLife*.

My most important concern should be relatively easy to address. It might seem to simply be a question of semantics, but I think it goes much more beyond than that. The title of the article is "Codon usage biases suppress premature transcription termination to promote gene expression". As formulated, it implies that there is actually an active mechanism that "senses" rare codons and feeds back this information to the transcription termination machinery. This would have been most intriguing and I was very curious to see what kind of mechanism the authors could propose for such a mechanism. But, in the core of the text, you understand that the most likely explanation is that codon bias on the one hand and transcription termination signal on the other hand coevolved to adapt to the G/C rich genome. In other words, there are no direct cause and effect relationship between codon bias and transcription termination, simply that both phenomenon, which are obviously linked to the same DNA sequence, co-evolved with it such as you could not change one without influencing the other. This is indeed the most likely explanation for these observations. Yet, in many instances, the way the text is written strongly implies that there is a direct mechanism that couples the two.

For examples: "These results suggest that codon usage plays an important role in mediating premature transcription termination[…]"; "[…]suggesting that codon usage bias also plays a role in premature transcription termination in mouse"; "[…]codon usage biases are a conserved mechanism that affects premature transcription termination[…]"; "[…]codon usage bias is a conserved mechanism[…]".

I strongly disagree with these formulations. Both, codon bias and transcription termination signals are correlated but, no, codon bias is not a "mechanism". These sentences, as well as the title, are very misleading (obviously provocative, but simply wrong) and the text should be re-written accordingly.

*Reviewer #3:*

The authors present evidence that shows that specific codons can trigger premature transcription termination in the organism Neurospora. They further analyse the sequence content of the 3'UTR polyA sites (PAS) and intragentic, what they call ORF-PAS and find slight differences in the PAS signals. They further show a strong anticorrelation of codon adaptation index and codon bias index with the ratio of ORF/3' termination. They do not find a similar mechanism in other fungi (*S. pombe*) yet in the mouse genome a similar albeit less significant correlation between codon usage and intragenic stop is observed. Another important observation was that the most frequent codon-pair was made up of two of the rarest codons and in fact resembles a canonical poly-A site.

Overall, the authors provide good evidence for a role in codon usage in transcription termination. However since these rare codons can make up a canonical polyA site, I would welcome some more analyses on codon pairs, and if it holds true that the mechanism of premature termination is not simply the formation of a PAS by two rare codons, it would be great if the authors speculate on the mechanism, e.g. by genome-wide analysis of ChIP-seq data (see comments below).

- It looks like the deoptimized codons to some extent act as stop codon, but not always. Also, when looking at pairs it seems it is easy to create a PAS, are the deoptimized codons introduced into the *frq* gene forming pairs that make up PAS?

- Does the premature stop in *NCU09435* and *NCU00931* (Figure 4) happen at non-optimal codons? Similarly, I'm missing a global analysis that counts the number of natural premature stop events per codon (rather than a correlation of indexes with ratios).

- Negative correlation of CAI and CBI with ratio of ORF-PAS vs. 3'PAS: what about the codon, or even codon-pair at the exact location of the premature stop? Are these predominantly non-optimal codons? Or is the mechanism rather additive and PolII needs to encounter a couple of non-optimal codons to stop?

- Figure 5: would be easier to read if it was on the same axis. Also the red line for K is somehow shifted. How does this correlation look for all AA within the same plot? Is it still visible or is it important to split them by AA?

- Is it just the formation of PAS that makes these rarer codons affect premature stop? I would welcome an analysis that looks at rare codons that do form a canonical PAS vs. those that don't and compare their effect on premature transcription stop. If it is mainly the formation of PAS that drives the termination I think the authors should be more explicit about this and say the mechanism of how rare codons affect premature termination is through forming non-canonical PAS.

- If it is not only the formation of a PAS (based on pair-wise analysis), what are the authors proposing as a mechanism? Is there any ChIP-seq data or similar available in Neurospora that shows an enrichment over the rare codons? For sure there is lots of ChIP-seq data available in mouse where the authors could look for factors potentially enriched at rare codons. Or does this coincide with PolII pausing?

- The authors have excluded an effect on chromatin based on looking at one mark in one locus. Is there any genome-wide data available that the authors could use for checking this statement across the naturally occurring rare codons? For sure in mouse there would be enough data for that.

[Editors' note: further revisions were requested prior to acceptance, as described below.]

Thank you for resubmitting your work entitled "Codon usage biases co-evolve with transcription termination machinery to suppress premature cleavage and polyadenylation" for further consideration at *eLife*. Your revised article has been favorably evaluated by Patricia Wittkopp (Senior Editor) and a Reviewing Editor.

The manuscript has been improved but there are some remaining issues that need to be addressed before acceptance, as outlined below:

You do not appear to have taken the *S. cerevisiae* literature on transcription termination into consideration when revising the manuscript. This will have to be done before final acceptance of the manuscript. If you do not deem this concern appropriate, then please provide an argument to this effect.

[Editors' note: further revisions were requested prior to acceptance, as described below.]

Thank you for resubmitting your work entitled "Codon usage biases co-evolve with transcription termination machinery to suppress premature cleavage and polyadenylation" for further consideration at *eLife*. Your revised article has been favorably evaluated by Patricia Wittkopp (Senior Editor) and a Reviewing Editor.

The manuscript has been improved but there are some remaining issues that need to be addressed before acceptance, as outlined below:

There appears to be a misunderstanding concerning the most recent editorial comment: "You do not appear to have taken the *S. cerevisiae* literature on transcription termination into consideration when revising the manuscript. This will have to be done before final acceptance of the manuscript. If you do not deem this concern appropriate, then please provide an argument to this effect.".

The referees of the original submission requested a referencing of the widely accepted link between codon bias and premature transcription termination by the Nrd1-Nab3-Sen1 dependent pathway in *S. cerevisiae*. Even though this is poly(A) site independent, conceptually this phenomenon is similar to the one described in the manuscript and hence deserves mentioning. The key reference is doi: 10.1093/nar/gkw683. Hence, either mention this literature or provide an argument why this is not relevant. Previously added references concerning *S. cerevisiae* polyadenylation do not appear relevant and may be deleted again.

---

## [Author Response]

[…] Major points:1) The title of the article is "Codon usage biases suppress premature transcription termination to promote gene expression". As formulated, it implies that there is actually an active mechanism that "senses" rare codons and feeds back this information to the transcription termination machinery. This would be very intriguing, but it appears that the most likely explanation is that codon bias on the one hand and transcription termination signal on the other hand co-evolved to adapt to a G/C rich genome. In other words, there is no direct cause and effect relationship between codon bias and transcription termination. Despite this likely explanation, the authors, in many instances, strongly indicate that there is a direct mechanism that couples the two phenomena. For example: "These results suggest that codon usage plays an important role in mediating premature transcription termination…"; "…suggesting that codon usage bias also plays a role in premature transcription termination in mouse"; "…codon usage biases are a conserved mechanism that affects premature transcription termination…"; "…codon usage bias is a conserved mechanism…".These formulations are way too strong. Codon bias and transcription termination signals are correlated but it is not a "mechanism". Thus, these sentences, as well as the title, are misleading (obviously provocative, but also simply wrong) and the title should be changed and the text re-written accordingly.

We agreed with this reviewer that codon usage, by affecting the formation of poly(A) signals, affects premature cleavage and polyadenylation. As suggested, we modified the title and the text in the revised manuscript.

2) A huge body of literature exists on premature transcription termination, which is ignored in the present manuscript. The findings should be discussed in the context of the relevant literature on (1) the link between codon bias and termination in S. cerevisiae and (2) premature termination at – typically intronic – premature cleavage and polyadenylation (PCPA) sites in mammals.

As suggested, we have now added more description and citations of previous studies on premature transcription termination in the revised introduction and discussion, especially on premature transcription termination in yeast and PCPA in introns of mammalian cells.

3) Since these rare codons can make up a canonical polyA site, more analyses on codon pairs would be welcome, and if it holds true that premature termination is not simply due to the formation of a PAS by two rare codons, it would be great if the authors could speculate on any other mechanism, e.g. by genome-wide analysis of ChIP-seq data. If it is not only the formation of a PAS (based on pair-wise analysis), what are the authors proposing as a mechanism? Is there any ChIP-seq data or similar available in Neurospora that shows an enrichment over the rare codons? For sure there is lots of ChIP-seq data available in mouse where the authors could look for factors potentially enriched at rare codons. Or does this coincide with PolII pausing? Moreover, the authors have excluded an effect on chromatin based on looking at one mark in one locus. Is there any genome-wide data available that the authors could use for checking this statement across the naturally occurring rare codons? For sure in mouse there would be enough data for that.

From our results and previous studies, it is clear that premature termination is not simply due to the formation of a single PAS motif. Other *cis*-elements surrounding the PAS motif are also very important for PCPA. Using the *frq* gene as an example in *Neurospora*, we showed that a cluster of rare synonymous codons can potentially form the A/U rich poly(A) signal, including PAS motif and other surrounding *cis*-elements, which could be recognized by transcription termination machinery. On the other hand, PCPA didn’t occur in the wild-type *NCU02034* gene even though a canonical PAS motif was present in the ORF (Figure 5—figure supplement 1). We showed that by introducing rare codons surrounding the PAS signal, transcription was fully terminated in the coding region (Figure 5).

To further address this issue, we performed genome-wide analyses in *Neurospora* and mouse by identifying “true PAS” and “false PAS” motifs based on the presence of poly(A) sites downstream of the PAS motifs and determined the nucleotide composition surrounding the groups of PAS motifs (Figure 5—figure supplement 2 and Figure 7—figure supplement 2). Our results clearly showed that in the ORFs, single PAS motif cannot trigger PCPA. In addition, the AU contents surrounding true PAS motifs are higher than in false PAS motifs, indicating that surrounding *cis*-elements are also required for triggering PCPA. These results are consistent with our results that the regions surrounding true PAS motifs are enriched for rare codons. Therefore, we propose that rare codons can potentially promote PCPA by the formation of PAS motif and its surrounding *cis*-elements.

As for the potential effect of chromatin on PCPA, we did the following analysis. We showed that the H3K9me3 levels at *frq* promoter were comparable in the wt-*frq* and *frq*-deopt2 strains (Figure 2—figure supplement 1), indicating that the loss of full-length *frq* mRNA in the *frq*-deopt2 strains is not due to H3K9me3-mediated transcriptional silencing. We have also done polII, H3K4me3, and H3K36me3 ChIP-seq previously in the lab and we found that polII, H3K4me3, and H3K36me3 enrichment positively correlates with codon usage and mRNA levels in *Neurospora*. We think this is mainly due to the effect of codon usage on transcription, which we are currently investigating to understand the mechanism.

As for the suggested ChIP-seq analysis to look for factors potentially enriched at rare codons, we attempted but we felt the results could not be easily interpreted because the resolution of ChIP-seq results (typically ~200 bp) does not have a codon-level resolution.

Reviewer #1:[…] Overall, the data shown are convincing and provide a conceptually new, though possibly not very surprising, link between codon bias and polyadenylation-dependent termination. The major caveat is that the findings are not discussed in the context of the relevant literature on (1) the link between codon bias and termination in S. cerevisiae and (2) premature termination at – typically intronic – premature cleavage and polyadenylation (PCPA) sites in mammals.

We appreciated the positive comments by the reviewers. As suggested, we have now added more description and citations of previous studies on premature transcription termination in the revised introduction and discussion, especially on premature transcription termination in yeast and PCPA in introns of mammalian cells.

Additional points:- (regarding deopt3): "suggesting that these products may be rapidly degraded" and "Because premature transcription termination products are usually rapidly degraded in the cytosol (van Hoof et al., 2002; Frischmeyer et al., 2002)". This point is probably of general relevance and should be discussed more explicitly. That is, the mostly low abundance of ORF PAS usage may well be due to the unstable nature of prematurely terminated transcripts. It is also worth noting that the prematurely terminated transcripts could well be subjected to nuclear decay as implicated in decay of human PCPA terminated RNAs, in addition to the non-stop decay suggested by the authors. This should be referenced together with a more general discussion of yeast and human premature termination systems.

We agreed that prematurely terminated transcripts may be degraded in the nucleus in addition to the non-stop decay in the cytosol. Studies on premature transcription termination in yeast and human are now cited and described in the revised paper.

- "The ratios were less than 10% for 95% of the genes with ORF-pA, indicating that only a small portion of transcription terminated in ORFs." The authors should discuss this more carefully. The RNA 3' ends do not mark transcription termination as RNA polymerase is present well beyond the PAS, as also found by the authors themselves. In addition, the low levels of ORF-pA may be partially due to the unstable nature of prematurely terminated transcripts.

We agreed that prematurely terminated transcripts may be inherently unstable.

This sentence is revised as: “The ratios were less than 10% for 95% of the genes with ORF-pA, which may be due to that these non-canonical poly(A) signals within ORFs are less efficient in promoting premature cleavage and polyadenylation (Berg et al., 2012; Guo et al., 2011) or that the prematurely terminated RNAs are unstable (van Hoof et al., 2002; Frischmeyer et al., 2002; Doma and Parker, 2007; Vanacova and Stefl, 2007).”

- Figure 2: Is this understood correct that the authors claim that the antisense gene is also prematurely terminated when the sense frq is codon deoptimized? This warrants further explanation. Where does the premature termination occur and do the mutation create PAS on the antisense? I wonder whether this is some kind of artefact. If no plausible explanation for the premature termination of the antisense gene can be found also this should be mentioned.

The antisense transcript *qrf* is also prematurely terminated when *frq* is de-optimized as shown in Figure 2 and Figure 2—figure supplement 1. As suggested, the poly(A) sites of *qrf* in the *frq*-deopt1 strain were determined by 3’ RACE as shown in Figure 2—figure supplement 1. It should be noticed that a PAS motif already exists in the wt-*frq* gene but does not lead to PCPA until surrounding codons were deoptimized. This is consistent with our finding that other surrounding *cis*-elements are also required for efficient premature termination.

- The authors find that ORF PAS are generally "weaker". Could this be a technical issue since those sites are generally also less abundant and hence less well defined? On a different note, the experimentally found pA events may be derived from alternative PAS-independent events. This could be discussed.

The reviewer is right that prematurely terminated transcripts are less abundant which make them more difficult to determine. Because of this reason, we determined the poly(A) sites using nuclear RNA.

We agreed that the experimentally found pA events might be derived from alternative PAS-independent events. As suggested, this is now discussed in the revised manuscript when we described the identification of poly(A) sites.

- In the case of Mus musculus data, it may be interesting to further compare to PAS motifs of intronic PCPA sites.

As suggested, the nucleotide profile and PAS motif of intronic PCPA sites are now described and are shown in Figure 7—figure supplement 1.

- Figure 5: The authors show that the AAUAAA element in NCU02034 is not used due to optimal codons in the vicinity. This may allow for a supporting bioinformatic analysis where the codon optimality surrounding all ORF AAUAAA (or perhaps the AAUGAA) motifs is compared for genes, which use those sites, compared to genes where they are not used.

Thank you for your suggestion. As suggested, we did the following analyses.

In the ORF and 3’ UTR of all *Neurospora* genes, we search for all the sequence contexts that could consist of a putative PAS motif (the top 40 most frequently used PAS motifs in *Neurospora* and the top2 in mouse), and divided them into two groups: “true PAS”, which have at least one pA site within the 5-35nt downstream region, and “false PAS”, which don’t have pA site. We compared the nucleotide composition between these two groups and found out that the AU contents of the true PASs were higher than that in the false PASs, including two U-rich elements and two A-rich motifs flanking PAS in *Neurospora* (Figure 5—figure supplement 2). Similar results were observed in mouse (Figure 7—figure supplement 2).

In addition, we found that more rare codons are used in the regions surrounding true PAS than that in the false PAS. Therefore, we conclude that the *cis*-elements surrounding PAS motifs are important for PCPA in coding regions.

- Figure 1: despite the confirmation of the absence of protein by the absence of mRNA (Figure 1), some information regarding the anti-FRQ antibody would be needed (these do not seem to be provided either in the previous references). Is this antibody homemade? Which part of the protein does it target?

The FRQ antibody was generated by using the full-length FRQ protein as antigen and the paper describing the method is now cited in the revised manuscript.

- Figure 3 proper Western blot quantification would require a serial dilution to estimate the protein concentration range in which the signal is linear (such a dilution do not require to be shown but should be mentioned in the Materials and methods).

Western blot quantification method was rewritten in the method.

- "other cis-elements such as U-rich upstream auxiliary element and U-rich downstream element also play important roles. In plants and yeast, the downstream element can be replaced by a U-rich element." These sentences are confusing and should be rephrased.

As suggested, we revised these sentences.

- Potential typo: "-30 ~ -10".

Thank you, the typo has been corrected.

- Discussion concerning the previous findings that codon usage affects chromatin structure and transcription initiation. For the reader it would be interesting to relate those earlier findings in the context of the premature termination. That is, is it possible that changes in chromatin structure and transcription levels are indirectly due to premature termination and vice versa?

We have done H3K4me3 and H3K36me3 ChIP-seq and we found that H3K4me3 and H3K36me3 are positively correlated both with CBI and mRNA levels in *Neurospora*. We think this is mainly due to the effect of codon usage on transcription, which we are currently working on to understand the underlying mechanisms.

It is possible that PCPA in coding region is affected by transcription level but we currently have no clear evidence to support this.

Reviewer #2:[…] My most important concern should be relatively easy to address. It might seem to simply be a question of semantics, but I think it goes much more beyond than that. The title of the article is "Codon usage biases suppress premature transcription termination to promote gene expression". As formulated, it implies that there is actually an active mechanism that "senses" rare codons and feeds back this information to the transcription termination machinery. This would have been most intriguing and I was very curious to see what kind of mechanism the authors could propose for such a mechanism. But, in the core of the text, you understand that the most likely explanation is that codon bias on the one hand and transcription termination signal on the other hand coevolved to adapt to the G/C rich genome. In other words, there are no direct cause and effect relationship between codon bias and transcription termination, simply that both phenomenon, which are obviously linked to the same DNA sequence, co-evolved with it such as you could not change one without influencing the other. This is indeed the most likely explanation for these observations. Yet, in many instances, the way the text is written strongly implies that there is a direct mechanism that couples the two.For examples: "These results suggest that codon usage plays an important role in mediating premature transcription termination[…]"; "[…]suggesting that codon usage bias also plays a role in premature transcription termination in mouse"; "[…]codon usage biases are a conserved mechanism that affects premature transcription termination[…]"; "[…]codon usage bias is a conserved mechanism[…]".I strongly disagree with these formulations. Both, codon bias and transcription termination signals are correlated but, no, codon bias is not a "mechanism". These sentences, as well as the title, are very misleading (obviously provocative, but simply wrong) and the text should be re-written accordingly.

Thanks for your suggestion. We agreed that codon usage, by affecting the formation of poly(A) signals, affects premature cleavage and polyadenylation. As suggested, we revised the title, the sentences listed here, and other sentences in the paper.

Reviewer #3:[…] Overall, the authors provide good evidence for a role in codon usage in transcription termination. However since these rare codons can make up a canonical polyA site, I would welcome some more analyses on codon pairs, and if it holds true that the mechanism of premature termination is not simply the formation of a PAS by two rare codons, it would be great if the authors speculate on the mechanism, e.g. by genome-wide analysis of ChIP-seq data (see comments below).

As suggested, more analysis on codon pair was performed. We found that premature transcription termination is not simply due to the formation of a PAS motif, other *cis*-elements are also very important for PCPA (Figure 5—figure supplement 2 and Figure 7—figure supplement 2).

- It looks like the deoptimized codons to some extent act as stop codon, but not always. Also, when looking at pairs it seems it is easy to create a PAS, are the deoptimized codons introduced into the frq gene forming pairs that make up PAS?

Codon de-optimization of *frq* resulted in a potential PAS (AAUAAU in *frq*-deopt1 and AAUAAA in *frq*-deopt2) which is located 18-nt upstream of poly(A) sites mapped by 3’ RACE (Figure 2).

- Does the premature stop in NCU09435 and NCU00931 (Figure 4) happen at non-optimal codons? Similarly, I'm missing a global analysis that counts the number of natural premature stop events per codon (rather than a correlation of indexes with ratios).

Transcription does not terminate at a single site but rather in a small region downstream of the PAS motif and forms a cluster of poly(A) sites. Therefore, it is difficult to calculate premature termination events at codon-level resolution.

As shown in Figure 4—figure supplement 1, premature termination sites are mapped to a small region of *NCU09435* and *NCU00931* ORF. And, both these two regions appear to be enriched in non-optimal codons.

- Negative correlation of CAI and CBI with ratio of ORF-PAS vs. 3'PAS: what about the codon, or even codon-pair at the exact location of the premature stop? Are these predominantly non-optimal codons? Or is the mechanism rather additive and PolII needs to encounter a couple of non-optimal codons to stop?

As mentioned above, transcription is not terminated at a single site but rather in a small region downstream of the PAS motif and forms a cluster. And it is difficult to determine at which codon or codon pair termination occurs. It is important to note that it is not codon or codon pair terminate transcription. Termination relies on the presence of the poly(A) signals, which include the PAS motif and its surrounding *cis*-elements. The use of clusters of rare codons will lead to the formation of potential poly(A) signals that terminate transcription.

- Figure 5: would be easier to read if it was on the same axis. Also the red line for K is somehow shifted. How does this correlation look for all AA within the same plot? Is it still visible or is it important to split them by AA?

As suggested, this figure was revised in current manuscript. As suggested, scatter plots showing the correlation between normalized codon usage frequency (NCUF) and relative synonymous codon adaptiveness of all codons with at least two synonymous codons are shown. Negative correlation was observed for both *Neurospora* (Figure 5 = -0.55) and mouse (Figure 7—figure supplement 2 = -0.36).

- Is it just the formation of PAS that makes these rarer codons affect premature stop? I would welcome an analysis that looks at rare codons that do form a canonical PAS vs. those that don't and compare their effect on premature transcription stop. If it is mainly the formation of PAS that drives the termination I think the authors should be more explicit about this and say the mechanism of how rare codons affect premature termination is through forming non-canonical PAS.

We found that premature transcription termination is not simply due to the formation of a PAS motif, other surrounding *cis*-elements are also very important for PCPA. Clusters of rare codons appear to form the A/U rich poly(A) signal, including PAS motif and other surrounding *cis*-elements, that can be recognized by transcription termination machinery.

We showed that PCPA didn’t occur in the wild-type *NCU02034*, even though a PAS motif was found in the wild-type gene (Figure 5—figure supplement 1). After codons surrounding the PAS was deoptimized, transcription was fully terminated in the coding region.

In addition, we performed genome-wide analyses in *Neurospora* and mouse by identifying “true PAS” and “false PAS” motifs based on the presence of poly(A) sites downstream of the PAS motifs and determined the nucleotide composition surrounding the groups of PAS motifs (Figure 5—figure supplement 2 and Figure 7—figure supplement 2). Our results clearly showed that in the ORFs, single PAS motif is not sufficient to trigger PCPA. In addition, the AU contents surrounding true PAS motifs are higher than in false PAS motifs, indicating that surrounding cis-elements are also required for triggering PCPA. These results are consistent with our results that the regions surrounding true PAS motifs are enriched for rare codons. Therefore, we propose that rare codons can potentially promote PCPA by the formation of PAS motif and its surrounding cis-elements.

- If it is not only the formation of a PAS (based on pair-wise analysis), what are the authors proposing as a mechanism? Is there any ChIP-seq data or similar available in Neurospora that shows an enrichment over the rare codons? For sure there is lots of ChIP-seq data available in mouse where the authors could look for factors potentially enriched at rare codons. Or does this coincide with PolII pausing?- The authors have excluded an effect on chromatin based on looking at one mark in one locus. Is there any genome-wide data available that the authors could use for checking this statement across the naturally occurring rare codons? For sure in mouse there would be enough data for that.

We have shown that H3K9me3 levels at *frq* promoter were comparable in the wt-*frq* and *frq*-deopt2 strains. This result indicates that the loss of full-length *frq* mRNA in the *frq*-deopt2 strains is not due to H3K9me3-mediated transcriptional silencing.

We have done polII, H3K4me3, and H3K36me3 ChIP-seq before and we found that polII, H3K4me3, and H3K36me3 are positively correlated with CBI and mRNA levels in Neurospora. We think this is mainly due to the effect of codon usage on transcription, which we are currently investigating to understand the underlying mechanism.

As for ChIP-seq analysis to look for factors potentially enriched at rare codons, we attempted but we felt the results could not be interpreted because the resolution of ChIP-seq results (typically ~200 bp) do not have a codon-level resolution.

[Editors' note: further revisions were requested prior to acceptance, as described below.]

The manuscript has been improved but there are some remaining issues that need to be addressed before acceptance, as outlined below:You do not appear to have taken the S. cerevisiae literature on transcription termination into consideration when revising the manuscript. This will have to be done before final acceptance of the manuscript. If you do not deem this concern appropriate, then please provide an argument to this effect.

To address the concern that there is a lack of *S. cerevisiae* literatures, we now added more than 10 yeast references in the revised paper:

Mischo and Proudfoot, 2013; Ozsolak et al., 2010; Moqtaderi et al., 2013; Mata, 2013; Schlackow et al., 2013; Vavasseur and Shi, 2014; Liu et al., 2017; Guo and Sherman, 1996; Graber et al., 1999; Graber, McAllister and Smith, 1999; Lemay et al., 2016; West and Proudfoot, 2009; Shalem et al., 2015; Lemay and Bachard, 2015; Larochelle, Hunyadkurti and Bachard, 2017.

[Editors' note: further revisions were requested prior to acceptance, as described below.]

The manuscript has been improved but there are some remaining issues that need to be addressed before acceptance, as outlined below:There appears to be a misunderstanding concerning the most recent editorial comment: "You do not appear to have taken the S. cerevisiae literature on transcription termination into consideration when revising the manuscript. This will have to be done before final acceptance of the manuscript. If you do not deem this concern appropriate, then please provide an argument to this effect.".The referees of the original submission requested a referencing of the widely accepted link between codon bias and premature transcription termination by the Nrd1-Nab3-Sen1 dependent pathway in S. cerevisiae. Even though this is poly(A) site independent, conceptually this phenomenon is similar to the one described in the manuscript and hence deserves mentioning. The key reference is doi: 10.1093/nar/gkw683. Hence, either mention this literature or provide an argument why this is not relevant. Previously added references concerning S. cerevisiae polyadenylation do not appear relevant and may be deleted again.

To address the concern you pointed out, we now added the suggested reference (Cakiroglu, Zaugg and Luscombe, 2016) and description of the study in the revised paper.